

# EnKF and 4D-Var Data Assimilation with a Chemistry Transport Model

Sergey Skachko[1], Richard Ménard[2], Quentin Errera[1], Yves Christophe[1], and Simon Chabrillat[1]

[1]Royal Belgian Institute for Space Aeronomy, BIRA-IASB, Brussels, Belgium
[2]Air Quality Research Division, Environment Canada, Dorval, Canada

*Correspondence to:* Sergey Skachko
(Sergey.Skachko@aeronomie.be)

**Abstract.** We compare two optimized chemical data assimilation systems, one based on the ensemble Kalman filter (EnKF) and the other based on four-dimensional variational (4D-Var), using a comprehensive stratospheric chemistry transport model (CTM). The work is an extension of the Belgian Assimilation System for Chemical ObsErvations (BASCOE), initially designed to work with a 4D-Var data assimilation. A strict comparison of both methods in the case of chemical tracer transport was done

in a previous study and indicated that both methods provide essentially similar results. In the present work, we assimilate observations of ozone, HCl, $HNO_3$, $H_2O$ and $N_2O$ from EOS Aura-MLS data into the BASCOE CTM with a full description of stratospheric chemistry. Two new issues related to the use of full chemistry model with EnKF are taken into account. One issue concerns to a large number of error variance parameters that need to be optimized. We estimate an observation error parameter as function of pressure level for each observed species using the Desroziers' method. For comparison reasons,

we apply the same estimate procedure in the 4D-Var data assimilation, where we keep both estimates: the background and observation error variances. However in EnKF, the background error covariance is modelled using the full chemistry model and a model error term. We found that it is adequate to have a single model error based on the chemical tracer formulation that is applied for all species. This is an indication that the main source of model error in chemical transport model is due to the transport. The second issue in EnKF with comprehensive atmospheric chemistry models is the sampling errors between species.

When species are weakly chemically related, cross-species sampling noise errors occur at the same location. These errors need to be filtered out, in addition to a localization based on distance. The performance of two data assimilation methods was assessed through an eight-month long assimilation of limb sounding observations from EOS Aura-MLS. The paper discusses the differences in results and their relation to stratospheric chemical processes. Generally speaking, EnKF and 4D-Var provide results of comparable quality but differ substantially in presence of model error or observation biases. If the erroneous chemical

modelling is associated with not too small chemical life-times, then EnKF performs better, while 4D-Var develops spurious increments in the chemically related species. If, on the other hand, the observation biases are significant, then 4D-Var is more robust and is able to reject erroneous observations, while EnKF does not.



## 1   Introduction

The Ensemble Kalman Filter (EnKF) and the four-Dimensional Variational algorithm (4D-Var) are widely used data assimilation methods that utilize the model to propagate observational information in time and space into an estimate of the state. Each method is built around different assumptions and has its own merits. But to some extent, the relative merits are application dependent. In the context of meteorological data assimilation, the relative advantages of these two methods have been discussed by Lorenc (2003) and Kalnay et al. (2007) to name a few, and it has promoted the development of new hybrid methods such as 4DEnVar (Lorenc et al., 2015) and En4DVar (Liu et al., 2008; Poterjoy and Zhang, 2015). In atmospheric chemistry there are, however, very few comparison studies. The purpose of this paper is to compare carefully optimized EnKF and 4D-Var chemical data assimilation systems for an extended time period using the same Chemistry-Transport Model (CTM) and same observations.

A short literature review discussing the Chemical Data Assimilation (CDA) problems related to EnKF and 4D-Var, their inter-comparison and application to the atmospheric chemistry modelling is already given in Skachko et al. (2014, hereafter denoted S14). More recent review including future prospects for coupled chemistry-meteorology models is given by Bocquet et al. (2015).

As in S14, here we use BASCOE (Belgian Assimilation System for Chemical ObsErvations) environment. BASCOE was designed to assimilate satellite observations of chemical composition into a stratospheric CTM originally using the 4D-Var assimilation method (Errera et al., 2008; Errera and Ménard, 2012). S14 described the implementation of the EnKF as an alternative assimilation method in BASCOE and compared it with the original 4D-Var approach, using carefully calibrated error variances for both methods and applying them to observations of ozone which was considered as a passive tracer. Indeed this preliminary paper performed the comparison in a chemical tracer transport framework, i.e. taking only transport into account while neglecting chemical reactions. Our results showed that in this framework the two methods give nearly identical performance. This outcome can be interpreted as a consequence of the dynamics of tracer error covariances: as noted early on by Cohn (1993) and Ménard and Daley (1996), such error covariances follow the characteristics of the flow. Hence in the absence of model error, there is thus no distinction between a filtering (EnKF) and a smoothing (4D-Var) algorithm.

But how do the EnKF and the 4DVar methods compare when photochemical reactions are taken into account? Do the results depend on the assimilated chemical species? Using actual satellite datasets and operational configurations, what are their respective performances in terms of precision, accuracy and computational efficiency? These are the main questions addressed in this paper.

The application of the multi-variate EnKF method to an assimilation system with the full chemistry should in principle address two important issues: the estimation of a large number of input error statistics; and the problem of localization between chemical species.

The first issue is the large number of input error statistics that is needed (e.g. the observation error variances for each species at each vertical levels). Clearly, an online estimation of error statistics is desirable to accomplish this task. In an idealized framework, Mitchell and Houtekamer (2000) proposed an adaptive EnKF where the model error parameters were



estimated using innovation statistics within a maximum likelihood method. In the same line of thought, the Desroziers' method (Desroziers et al., 2005) was also used to simultaneously estimate the covariance inflation and the observation errors (Li et al., 2009; Gaubert et al., 2014). Ménard (2016) also showed that the Desroziers' observation variance estimates converge to the truth if the background error is close to the truth, which seems to be a reasonable assumption for EnKF background error

covariances, when the $\chi^2$ condition (Ménard and Chang, 2000) is respected.

     The second issue related to the implementation of a multi-species EnKF is the localization between species. It is well known in EnKF applications that a tapering of the sampling error correlations is needed when the true error correlation is not close to +1 or -1 (e.g. Anderson, 2012). For correlations that depend on distance, a widely used sampling error correction is provided by the Schur product of a compact support correlation function (Gaspari and Cohn, 1999) to the sample covariance. However,

in comprehensive atmospheric chemistry models that have many prognostic chemical species, sampling errors between species at the same location are also expected to occur. The approach to this cross-species sampling correlation noise has not been fully explored yet. Several studies in EnKF chemical data assimilation use a brute force species localization that consists in zeroing-out cross-species covariances. This is the case for example for Tang et al. (2011) and Gaubert et al. (2014) where only $O_3$ is observed and where all cross-covariances between ozone and other species are zeroed-out in order to reduce the noise

in the analysis. In an ozone assimilation study, Curier et al. (2012) kept the cross-covariances between $O_3$ and some other strongly coupled species, in particular NO, $NO_2$ and VOC's, as well as the error covariances with the boundary conditions ($O_3$ dry deposition and model top boundary condition). They showed that each of these kept cross-covariances give rather similar impact on the ozone analysis. Eben et al. (2005) in an multi-species air quality EnKF assimilation of surface $O_3$, NO and $NO_2$ measurements indicated that in order to reduce the sampling noise they kept the cross-covariances between these

species only at the surface. In another study, Miyazaki et al. (2012) assimilated simultaneously $NO_2$, $O_3$ CO, and $HNO_3$ tropospheric chemical species along with the estimation of surface emissions. Using verification against satellite observations, such as innovation variance, they found that cross-covariance between chemical species need to be set to zero unless they are strongly chemically related. Examples of strongly chemically related species are members of the NOy family, or CO with VOC's. Miyazaki et al. (2012) also allowed the coupling between $NO_2$ and emissions of $NO_2$, or the CO with the emissions of

CO, but set to zero the cross-covariance between emissions of $NO_x$ and CO. Keeping the cross-covariance with the boundary conditions (surface emissions and lateral boundary conditions) was also argued in Constantinescu et al. (2007). Overall, these studies indicate that when there is a believed strong correlation between observed and modelled species (or boundary condition) then these can be kept in the EnKF, but otherwise to reduce noise, all other cross-covariances are better be zeroed-out.

     In this paper we perform an assimilation with EnKF and 4D-Var of several species in the stratosphere that are not necessarily

directly chemically linked and with real-life constraints. The lifetimes of the assimilated species are quite diversified and vary with altitude. We use a state-of-the-art CTM that is in fact in constant improvement, but also has some deficiencies. We use limb sounding observations that give vertically resolved measurements, and thus there is a need to have vertically resolved error statistics. As it was shown in S14, the EnKF is more sensitive to the observation error statistics than 4D-Var assimilation. Yet, to provide a consistency between the two assimilation systems, the observation error statistics of 4D-Var will be a subject





to the same Desroziers' estimation procedure. Localization between species, that is needed in EnKF, is in fact not applied to 4D-Var, since the cross-covariance between species are taken into account automatically using the 4D-Var adjoint model.

The paper is organized as follows. The next section describes the main components of the BASCOE Data Assimilation System (version 5.8): the common CTM, the 4D-Var system and the EnKF system. It also describes the implementation of

Desroziers' method and the tuning of the error covariances in each system. The assimilated observations and independent data used to validate the results are given in Sect. 3. Section 4 describes the results of our assimilation and model experiments. And Sect. 5 discusses a separate EnKF experiment where the cross-species correlations are taken into account. Finally, some conclusions are given in Sect. 6.

## 2   The BASCOE Data Assimilation System

### 2.1   The Chemistry-Transport Model

In this study, all numerical experiments are performed with the Belgian Assimilation System for Chemical ObsErvations (BASCOE) and its underlying Chemistry Transport Model (CTM). The BASCOE CTM computes the temporal evolution of 58 stratospheric chemical species accounting for the advection, photochemical reactions and a parametrization of PSC microphysics. We used a CTM configuration nearly identical to the one described by (Lefever et al., 2015) for Near Real-Time

production of 4D-Var analyses as part of the MACC project. Here we provide a brief reminder of its most salient features.

All species are advected by the Flux-Form Semi-Lagrangian scheme (Lin and Rood, 1996), here driven by ERA-Interim wind fields (Dee et al., 2011). The horizontal resolution is set at $3.75°$ longitude by $2.5°$ latitude. The model considers 37 levels from the surface to 0.1 hPa, which is a subset of the 60 levels of ERA-Interim that excludes most tropospheric levels. Hence The CTM state is described by the vector $\mathbf{x} \in \mathfrak{R}^n$ of length $n = 96 \times 73 \times 37 \times 58 \approx 1.5 \times 10^7$. The model time step is

set to 30 minutes.

The photochemical scheme of BASCOE account for 208 stratospheric chemical reactions: 146 gas-phase, 53 photolysis, 9 heterogeneous. Photolysis rates are provided by the Jet Propulsion Laboratory (JPL) recommendations (Sander et al., 2006). The computation of the photolysis rates is based on the Tropospheric Ultraviolet and Visible (TUV) radiative transfer package (Madronich and Flocke, 1999).

### 2.2   Setting up the time windows

In order to describe the practical implementations of the 4D-Var and EnKF algorithms in BASCOE, we must first explain the different set-up of their assimilation windows with respect to time. This is schematically shown by Fig. 1. The 4D-Var assimilation window is set to 24 h, i.e. this is the duration of the forward and backward integrations of the CTM and its adjoint. Each 4D-Var iteration is followed by a minimizing procedure (see Sect. 2.3 for more details). In this 4D-Var implementation, the

24 h forecast is defined as the first forward model simulation starting from the analysis of the previous assimilation window. All





4D-Var assimilation cycles save the model state in observation space during these forecasts, in order to compute Observation-minus-Forecast (OmF) statistics discussed below.

The EnKF initializes its ensemble of model states from one given state using a procedure described in Sect. 2.4. The EnKF assimilation is then based on ensembles of short model forecasts which have the same duration as the CTM time step, i.e. 30

minutes, followed by the observational update of each ensemble member. The updated ensemble states (analyses) are used then as initial states for the next ensemble forecast. Hence, there is no practical need to compute the 24 h forecast (green line) as in the 4D-Var approach. However we have introduced this option in the EnKF in order to allow a consistent comparison with the 4D-Var forecasts. Hence in the EnKF approach, the 24 h forecast is defined as a model simulation started from the ensemble mean analysis at 0 h UTC. As in the 4D-Var system, the 24 h forecast of the EnKF stores the OmF statistics.

## 2.3  The 4D-Var system

The BASCOE 4D-Var of this study was already used by S14 and is described in detail by Errera and Ménard (2012). The 4D-Var data assimilation is carried out by minimizing the so-called cost function which measures the discrepancy between the model state and observations (Talagrand and Courtier, 1987). Here, the model state vector contains 58 prognostic variables, where only 7 chemical species are observed among them, see Sect. 3.

The background error covariance matrix $\mathbf{B}_0$ is parametrized using a control variable transform

$$\mathbf{L}\xi = \mathbf{x}_0 - \mathbf{x}_0^b \equiv \delta\mathbf{x}_0, \tag{1}$$

where $\xi$ is a new control variable, $\mathbf{x}_0$ the first guess field, $\delta\mathbf{x}_0$ is the analysis increment and $\mathbf{L}$ is the square root of $\mathbf{B}_0$:

$$\mathbf{B}_0 = \mathbf{L}^T\mathbf{L}. \tag{2}$$

As in S14, the error covariance of the first guess field expresses spatial correlations on a spherical harmonic basis (Courtier

et al., 1998), allowing a representation of homogeneous and isotropic horizontal correlations by a diagonal matrix with diagonal values repeated for the same zonal wave number. The operator $\mathbf{L}$ is defined by

$$\mathbf{L} = \boldsymbol{\Sigma}\mathbf{S}\boldsymbol{\Lambda}^{1/2}, \tag{3}$$

where $\boldsymbol{\Sigma}$ is the (diagonal) background error standard deviation matrix with $s_b(l)\sigma_b(l)$ values on its diagonal, $s_b(l)$ is an *adjustable background error scaling factor* on the level $l$; $\boldsymbol{\Lambda}^{1/2}$ is the spatial correlation matrix defined on a spherical harmonic

basis hence diagonal; and $\mathbf{S}$ is the spectral transform operator from the spectral space to the model space. The spatial correlation matrix considers Gaussian correlations in the horizontal and in the vertical directions with length scales $L_0^h$ and $L_0^v$ in horizontal and vertical directions, respectively.

The observation errors are assumed to be uncorrelated both horizontally and vertically. The observation error covariance matrix $\mathbf{R}_k$ is thus defined diagonal:

$$\mathbf{R}_k(i,j) = \begin{cases} (s_o(i)\,\sigma_y(i)|_{t_k})^2, & \text{if } i = j \\ 0, & \text{if } i \neq j, \end{cases} \tag{4}$$





where $s_o(i)$ is an *adjustable observation error scaling factor* and $\sigma_y(i)|_{t_k}$ is the measurement error at level $i$ and time $t_k$. The observations and their errors are described in Sect. 3. The adjustment of $s_b$ and $s_o$ scaling factors is performed in observation space for every observed species separately, where they are functions of vertical pressure level (see Sect. 2.5).

Finally, the BASCOE 4D-Var implementation has the background quality control procedure (BgQC, Anderson and Järvinen, 1999).
This procedure rejects observations when:

$$(\mathbf{y}_{i,l} - H_{i,l}(\mathbf{x}_b))^2 > \gamma(\text{diag}(\mathbf{R})_{i,l} + H_{i,l}(\text{diag}(\mathbf{B}))) \tag{5}$$

where the operator $\text{diag}(\mathbf{A})$ is a diagonal matrix of $\mathbf{A}$ and $i,l$ are the data indices of profile and level, respectively. The value of $\gamma$ is set to 5 in BASCOE, so that BgQC rejects only obviously wrong observations.

## 2.4 The EnKF system

The BASCOE EnKF of this study is similar to the system used in S14. An ensemble of initial states $\tilde{\mathbf{x}}_i(t_0)$ is generated by adding to the model state $\mathbf{x}_0$ a set of spatially correlated perturbations according to the prescribed initial error covariance. This procedure is schematically represented on Fig. 1 on the left-hand side. The ensemble of model states is propagated forward in time using the same CTM as used in the 4D-Var (see Sect. 2.1). The model error covariance is represented by the addition of a stochastic noise $\boldsymbol{\eta}_i$ to each ensemble member at each model time step. In the current implementation, the model error term is added to observed species only. The non-observed model species evolve with ensemble and are influenced by the analysis increments only implicitly through the chemistry scheme of BASCOE CTM.

The operator $\mathbf{L}$ described in Sect. 2.3 is used to generate the initial deviation $\tilde{\mathbf{x}}_i(t_0)$ and the model error $\boldsymbol{\eta}_i(t_k)$ of the EnKF system. This ensures that at the initial time, both EnKF and 4D-Var systems have identical error statistics. Initial deviation is defined as

$$\tilde{\mathbf{x}}_i(t_0) = \mathbf{L}\boldsymbol{\zeta}_i(t_0), \quad i \in [1, N], \tag{6}$$

whereas the model error term is written as

$$\boldsymbol{\eta}_i(t_k) = \alpha \mathbf{L}\boldsymbol{\psi}_i(t_k), \quad i \in [1, N], \tag{7}$$

where $\boldsymbol{\zeta}_i(t_0)$ and $\boldsymbol{\psi}_i(t_k)$ are normally distributed random numbers with zero mean and variance equal to 1, defined in the spectral space; and where $\alpha$ is an empirical model error parameter. The definition of this parameter is explained in Sect. 2.6.

The observation error covariance matrix $\mathbf{R}$ is defined by Eq. 4, where the adjustable scaling factor $s_o(i)$ is also estimated using the Desroziers' method described in Sect. 2.5. The fact that the matrix $\mathbf{R}$ is calibrated automatically without using a trial and error procedure for every observed species makes EnKF essentially easier to parametrize than in our previous study. Besides, the Desroziers' methods allows for more accurate observation error estimation because it computes $s_o(i)$ as a function of observation vertical pressure level. It should be noted that EnKF uses the same background quality control procedure described in Sect. 2.3.





As in our previous study with a chemical tracer model, BASCOE EnKF uses the Schur (element-wise) product of the ensemble covariance matrix with a compact support correlation function. This function is the 5th-order piecewise rational function of Gaspari and Cohn (1999) which is isotropic and decreases monotonically with distance depending on the correlation length scale $L_{loc}$. The function is positive only for distances that are less than $2L_{loc}$ and zero otherwise. We applied this

procedure to both horizontal and vertical correlations, using the compact support correlation functions with correlation length scales $L_{loc}^h$ and $L_{loc}^v$, respectively. The choice of these parameters is discussed in Sect. 2.6. In order to make feasible the computation of much more expensive EnKF in the framework of full-chemistry model, the analyses are computed locally, around the area where current satellite observations are situated. To this end, the EnKF algorithm accounts now for a new effective procedure to find a current local sub domain in the model space.

The EnKF analyses of this study are performed in parallel for every observed species in its own space. Thus, such analysis increments of every species do not account for cross-correlations between different chemical observations, which is not the case for the 4D-Var system. However, it is technically possible to keep all observations from multiple species in one observation space, introducing thus the cross correlations between species. An example of such EnKF data assimilation is discussed in Sect. 5.

**2.5   The Desroziers' method**

We use the Desroziers et al. (2005) method to estimate error variance scaling factors for each observed species and each vertical levels. The diagnosis relies on linear estimation theory where the statistics is computed using observation-minus-background, observations-minus-analysis, and analysis-minus-background differences. The estimation of the background error variance is written as

$$s_b(l)^2 \sigma_b(l)^2 = \langle (\mathbf{d}_b^a)^T \mathbf{d}_b^o \rangle, \tag{8}$$

and the observation error variance is then

$$s_o(l)^2 \sigma_o(l)^2 = \langle (\mathbf{d}_a^o)^T \mathbf{d}_b^o \rangle, \tag{9}$$

where the vector $\mathbf{d}_b^a$ is the difference between analysis and background, $\mathbf{d}_b^o$, the difference between observations and background, $\mathbf{d}_a^o$, the difference between observations and analysis in observation space, and $\langle \rangle$ denote the mathematical expectation.

Note that in practical implementation, the expectation is replaced by a horizontal mean and time mean of a day.

The BASCOE data assimilation is initialized using $s_b(l) = 1$ and $s_b(l) = 1$ for both, EnKF and 4D-Var. These initial values are kept in the system for the first 24h of system integration. The analysis increment and model innovation statistics are accumulated during this time. Then the estimation of the scaling factors is performed using expressions 8 and 9. The following 24h analyses (on day 2), both EnKF and 4D-Var use the day 1 estimated error variance scaling factors. The procedure is

sequentially updated every 24h assimilation cycle using the statistics accumulated during the previous cycle. Note that we could have used an estimated $s_b(l)$ in EnKF to tune the model error, but we decided not to apply it and use the $\chi^2$ tendency to this end (see next section, Sect. 2.6).



The Desroziers' estimates appear to be asymptotically stable after only one day. That means that changing the initial parameter value has little to no effect on the resulting time series of estimated parameter values. Figure 2, shows examples of the observation error variance for $O_3$, $H_2O$ and HCl (at each vertical levels) when we perturb the initial parameter value by a factor 1.5. The dashed curves represent the initial values and the solid curves the values estimated after one day using the Desroziers' method.

## 2.6 Tuning of error covariances in the two systems

Each assimilation system (i.e. EnKF and 4D-Var) has its own optimized error variances but shares a common error correlation for the prescribed $\mathbf{B}_0$ in 4D-Var and the prescribed model and initial condition error correlations in EnKF. As in S14, our starting point is the calibration of the error covariance matrix $\mathbf{B}_0$ used by the 4D-Var system. This is realized through a calibration of the spatial correlation associated with the operator $\mathbf{L}$ described in Sect. 2.3. The operator $\mathbf{L}$ has similar parameters as in S14: $L_0^h = 800$ km and $L_0^v = 1$ model level, $\sigma_b(l) = 0.2$ (with scaling factor $s_b(l) = 1$ ) at all levels. Then, we take into account the fact that the use of the Schur product results in shorter correlation length scales (See S14 for more details). Similarly to our previous work, EnKF uses an effective correlation length scales of $L_e^h = 872$ km and $L_e^v = 1.3$ in model level coordinates, given that $L_{loc}^h = 2000$ km and $L_{loc}^v = 1.5$ are chosen a priori. The calibrated operator $\mathbf{L}$ is then used in the EnKF system.

The observation error scaling factor $s_o$ (see Sect. 2.4) is estimated for both systems using the Desroziers' method described in Sect. 2.5. The background error scaling factor $s_o$ used in 4D-Var is also estimated using the Desroziers' method. In EnKF, the background error covariance is evolved using CTM where we add a model error term that uses a calibration parameter $\alpha$. The value of $\alpha$ equal to 0.025 was found in S14 in the case study of $O_3$ tracer. This value is based on the property that the time tendency (over periods of weeks and months) of the $\chi^2$ diagnostic should be nearly zero as argued in Ménard and Chang (2000). In general, we have found (in S14) that the value of $\alpha$ changes the slope of the $O_3$ $\chi^2$ distribution, whereas the observation error scaling factor $s_o$ is responsible for the mean value of it. In the absence of better knowledge, we use the value $\alpha = 0.025$ for all observed species described in Sect. 3.

The performance of each data assimilation system of BASCOE can be monitored by the $\chi^2$ diagnostic. During the whole period of our experiments, $\langle \chi_k^2 \rangle / m_k$ values remain close to 1 (result not shown). This is achieved by using the error variances estimated by the Desroziers' method. In the case of 4D-Var where both observation and the background error variances are estimated, the Desroziers' method gives estimates that achieve the innovation variance consistency (Ménard, 2016). For EnKF where only the observation error is estimated, the fact that $\langle \chi_k^2 \rangle / m_k$ values remain close to 1 is an indirect confirmation that the model error is tuned appropriately. Figure 3 shows the evolution of the adjustable parameters for both systems. The solid lines show the vertically mean values of the observation variance parameter $s_o$ and the dashed lines, the vertically mean values of the 4D-Var background error variances. The time evolution of the error variance scaling factors at individual levels is, as in Figure 3, (result not shown) generally consistent over time, especially for the EnKF estimates, and thus goes along in arguing that they represent some true error statistics. Furthermore, we note that for the EnKF results, the scaling factors of any species show no drift in time. We argue from this result that there is apparently no need to have a different model error $\alpha$ for different





species. Thus we conclude that for a chemical transport models, the main source of model error can be attributed to transport errors primarily.

Finally, we wish to remark that to keep comparable CPU costs in both data assimilation systems, and that can be carried out in a reasonable time, 4D-Var is run with 10 iterations (including 10 adjoint iterations) and the EnKF uses 20 ensemble members. As in S14, the computation of the EnKF Kalman gain is performed using Cholesky decomposition where the full observation vector is considered at a given time step for a given species. No simplification is used to compute the inversion of the innovation matrix $[\mathbf{HB}_e\mathbf{H}^T + \mathbf{R}]$ or the matrix $\mathbf{B}_e\mathbf{H}^T$ (see S14 for more details). The actual use of local domain decomposition and integration of the ensemble members on different processors in parallel decreases essentially the CPU costs as compared to the previous version of BASCOE EnKF.

## 3 Observations

The dataset assimilated in this study is the version 4.2 of the retrievals from the Microwave Limb Sounder (MLS) on-board the EOS (Earth Observing System) Aura satellite (Livesey et al., 2015). Here we assimilate the retrievals of five species which are listed in Table 1 along with some key parameters of the dataset and the validation reference for each species.

Some results of the data assimilation will be validated against independent observations. This will be the case for $N_2O$ because the Aura MLS $N_2O$ precision is 24-14 ppbv (9-38%) and its accuracy is 70-3 ppbv (9-25%) in the pressure range 100-4.6 hPa but its precision drops to 14 ppbv (250%) at 1 hPa, where the accuracy is estimated to 16% (Livesey et al., 2015). Hence we will validate the BASCOE $N_2O$. with observations retrieved from the Atmospheric Chemistry Experiment - Fourier Transform Spectrometer (ACE-FTS) satellite instrument (Bernath et al., 2005) which uses solar occultation to provide around 28 profiles per day. Strong et al. (2008) validated $N_2O$ retrievals from ACE-FTS (version 2.2) and found a bias of $\pm 15\%$ between 6-30 km and a bias of $\pm 4$ ppbv between 30-60 km . Here we use the retrieval version 3.5.

We will also use the Michelson Interferometer for Passive Atmospheric Sounding (MIPAS) retrievals by the IMK/IAA (Institut für Meteorologie und Klimaforschung, Karlsruhe/Instituto de Astrofisica de Andalucia, Grenada) to validate the unconstrained distributions of $CH_4$ and $NO_x$ ($NO_x$= NO + $NO_2$). The MIPAS IMK/IAA retrievals of $CH_4$ were validated by Laeng et al. (2015) and the retrievals of $NO_x$ were described by Funke et al. (2005).

## 4 Numerical experiments

This section reports the numerical experiments performed in this study: the control run, i.e. an unconstrained simulation by the BASCOE CTM including photochemistry; the "EnKF" and "EnKF tracer" experiments, the first one including photochemistry and the second one neglecting it (i.e. assimilation in chemical tracer mode as done in S14); and the two corresponding "4D-Var" and "4D-Var tracer" experiments. All experiments start on 1 April 2008 from the same initial condition, i.e. a 4D-Var analysis of Aura-MLS retrievals (Lefever et al., 2015), and end on 1 November 2008 i.e. after 7 months.



**Table 1.** List of species retrieved in Aura-MLS v4.2 and assimilated for this paper.

| Name | Resolution (km) Vertical x Horizontal | Vertical range of assimilation (hPa) | Accuracy | Precision | Validation paper (for Aura-MLS v2.2) |
|---|---|---|---|---|---|
| $O_3$ | 3 - 6 x 200 - 300 | 0.1 - 261 | 3 - 20% | 2 - 40% | Froidevaux et al. (2008a) |
| $N_2O$ | 4 - 6 x 300 - 620 | 4.64 (1) - 100 | 9 - 25% (12%) | 7 - 38% (250%) | Lambert et al. (2007) |
| $H_2O$ | 3 - 4 x 220 - 440 | 0.1 - 316 | 4 - 11% | 4 - 9% | Lambert et al. (2007) |
| $HNO_3$ | 3 - 5 x 300 - 500 | 1.5 - 215 | ±0.5 - 2 ppbv | ±0.7 ppbv | Santee et al. (2007) |
| HCl | 3 - 6 x 200 - 400 | 0.32 - 100 | 5 -50% | 10 - 50% | Froidevaux et al. (2008b) |

The results of our model and data assimilation experiments will be assessed using Observations-minus-Forecast (OmF) statistics, relative bias and standard deviation, computed in the observation space. In the case of $N_2O$ the relative bias and standard deviation are not good diagnostics because its volume mixing ratio decreases by two orders of magnitude between 100 and 1 hPa. Hence we will simply compare the mean profiles of $N_2O$ by the five numerical experiments with assimilated

and independent measurements. The statistics are computed in three different latitude bands covering the globe: South Pole (90°S-60°S), middle latitudes and Tropics (60°S-60°N) and North Pole (60°N-90°N). The analyses of the assimilated species are verified by comparison with the assimilated observations (sections 4.1–4.5). Section 4.6 evaluates the results for methane and nitrogen oxides which are not assimilated.

## 4.1   Verification of ozone

Figure 4 shows the OmF statistics for ozone over the period September 2008-October 2008, i.e. during the period of the Antarctic ozone hole. The results of the tracer experiments are not shown above 1 hPa because the tracer approximation is not valid in this region. The CTM experiment delivers rather large biases (10 to 30%) in the lower and upper stratosphere and at all levels above the South Pole region.

All data assimilation experiments succeed in eliminating these biases nearly completely in the lower and middle strato-
sphere. The resulting biases are smaller than 2%, except for the 4D-Var experiment which overestimates ozone depletion in the Antarctic ozone hole region (around 50 hPa) by up to 5%. Compared with the CTM results, the 4D-Var and EnKF experiments also reduce significantly the OmF standard deviation in the lowest levels. The smallest OmF standard deviations are delivered by the 4D-Var experiment, with results about 1% smaller than those delivered by the EnKF in pressure range 30-2 hPa.

The experiments 4D-Var tracer and EnKF tracer allow us to assess the impact of stratospheric chemistry. Neglecting this
process results in larger biases and OmF standard deviations above the South Pole in the region 10-2 hPa, where both tracer data assimilation systems overestimate ozone by ~5% and deliver OmF standard deviations reaching 10%.

The photochemical lifetime of ozone decreases rapidly in the upper stratosphere and reaches values as short as a few minutes (Brasseur and Solomon, 2005) in the lower mesosphere. In these regions, our CTM experiment has a significant ozone deficit reaching about 20% at the stratopause (1 hPa). The sources of this model bias are out of scope of the present paper. However,





its presence helps to assess the behaviour of our assimilation algorithms. It is found that both data assimilation algorithms fail to correct this model bias: ozone is still underestimated by ∼15% at the Stratopause. In the upper stratosphere and mesosphere, data assimilation does not improve OmF standard deviations either: these remain nearly identical to those obtained by the CTM. These results indicate that when the photochemical lifetime is short (e.g. smaller than the time step of the model) and

the model error is important, both data assimilation systems fail to improve the representation of the model state. Since this issue also involves species that have strong chemical interactions with ozone, it will be further discussed in Sections 4.2, 4.4 and 4.6.

## 4.2   Verification of HCl

During the largest part of the CTM simulation, the HCl distribution is in agreement with the Aura-MLS observations. And

the EnKF and 4D-Var experiments deliver nearly identical results where the small CTM biases are completely corrected (not shown). The only exception is in the South Pole latitude band, during the period May-June 2008 which is shown on figure 5. During this period the chemical lifetime of HCl is much shorter than at other latitudes, because the heterogeneous removal due to the formation of Polar Stratospheric Clouds has already started. This loss process is currently overestimated in the BASCOE CTM, due to a crude cold-point temperature parametrisation (section 2.2.2 in Lefever et al., 2015). As a result, the CTM

experiment underestimates HCl by up to 45% at 30 hPa in the Antarctic polar vortex region and its OmF standard deviation also reaches ∼ 45% . While the 4D-Var approach essentially fails to correct this large disagreement, the bias is nearly halved in the EnKF experiment and the OmF standard deviation is significantly reduced as well.

Staying in the lower stratosphere (100-10 hPa), the outcome of the experiments is different than above the South Pole. Northward of $60°$S, the CTM biases do not exceed 15% and they are nearly eliminated by both data assimilation experiments.

The OmF standard deviations of both data assimilations are also quite similar in these regions.

In the middle stratosphere, the chemical lifetime of HCl decreases from about one week at 10 hPa to about one day at 1 hPa (Brasseur and Solomon, 2005). The CTM experiment delivers quite accurate results in this region: the OmF biases do not exceed 3% and the standard deviations are less than 10%, in every latitude band for the pressure range 10-0.46 hPa. The EnKF and 4D-Var experiments both succeed in correcting these small CTM biases and reducing the OmF standard deviations, except

at the 1 hPa level where the 4D-Var does not correct the CTM deficit of 3% for HCl.

## 4.3   Verification of HNO$_3$

Figure 6 shows the HNO$_3$ OmF statistics between the assimilated Aura MLS data and the CTM, EnKF and 4D-Var experiments for the period September-October 2008. For all three latitude bands, the CTM shows a significant underestimation reaching 20-25% around 30 hPa. This model bias nearly disappears at 10 hPa but grows again above this level. In the lower stratosphere,

the OmF standard deviations of the CTM experiment reach minimum values of 10-15% but at the lowermost levels the standard deviation is much larger in the Antarctic polar vortex region than at other latitudes.

Both data assimilation experiments correct the OmF model bias at all latitudes and at all pressure levels between 100 and 10 hPa. Above that level, the quickly increasing model OmF bias is not corrected by either assimilation algorithm. The





explanation for this different behaviour in the upper stratosphere is twofold. First, the observation error grows quickly with altitude, reducing the weight of observations in the assimilation experiments. Second, a large discrepancy between the model and the observed data leads to rejection of most measurements above 10 hPa by the background quality control procedure (see Sect. 2.3 for more details).

The 4D-Var OmF bias is generally less than 3% in the pressure range 100-7 hPa, except for an 8% OmF bias at 70 hPa in the Tropics. The EnKF delivers even smaller OmF biases in the whole pressure range and at all latitudes. Both data assimilations results in almost identical OmF standard deviations, except in the Antarctic polar vortex region where the EnKF errors are slightly larger below 20 hPa.

## 4.4   Verification of water vapour

Water vapour is a long-lived tracer in the whole stratosphere, with a photochemical lifetime still longer than one month at the stratopause (Brasseur and Solomon, 2005). The OmF statistics for $H_2O$ are shown on Fig. 7. The CTM provides OmF biases smaller than 10% in the whole pressure range and at all latitudes, except in the Antarctic polar vortex between 100 and 10 hPa, where $H_2O$ underestimation reaches 30%. The OmF standard deviation by the CTM is also largest in this region, reaching 23% while it does not exceed 15% elsewhere.

Both data assimilations mostly correct the OmF bias and standard deviation errors with respect to the CTM. Their OmF biases do not exceed 2%, except for the OmF bias by the 4D-Var which reaches 3% at 1 hPa, i.e. the level where the ozone deficit described in Sect. 4.1 is maximum. The OmF standard deviation errors resulting from the two assimilation experiments are also quite similar, with slightly larger EnKF errors in the Antarctic polar vortex below 10 hPa.

## 4.5   Verification of $N_2O$

The relative error statistics shown for other species are difficult to interpret in the case of $N_2O$ because its volume mixing ratio decreases by two orders of magnitude between 100 and 1 hPa. Hence figure 8 simply compares mean profiles of forecasts and observations. We display the assimilated Aura MLS observations along with their validation uncertainties (grey filled region as reported by Lambert et al., 2007). Since these uncertainties are very large in the upper stratosphere, we also compare with independent observations by the ACE-FTS solar occultation instrument (see Sect. 3).

In the lower stratosphere the two satellite datasets and the CTM experiment are in good agreement. Above 10 hPa the mixing ratios retrieved from Aura-MLS are much larger than those from ACE-FTS, and above 5 hPa they become pressure-independent which is not realistic. As expected, the CTM experiment agrees much better with the ACE-FTS $N_2O$ retrievals since they are much more precise in the upper stratosphere.

How do the 4D-Var and EnKF treat the tricky Aura-MLS dataset? To answer this question we inhibited any a priori filtering
of the Aura MLS observations of $N_2O$ above 5 hPa, and we used both the full chemistry CTM and its transport-only version. Figure 8 shows that both EnKF experiments follow the assimilated Aura MLS data in the upper stratosphere, whereas the mean profile delivered by the 4D-Var experiment remains closer to the CTM. This is due to the automatic rejection by the 4D-Var of most Aura-MLS observations of $N_2O$ above 5 hPa However, 4D-Var assimilation with a chemical tracer transport model (cyan



dashed curve) assimilates more Aura MLS data approaching closer to them. This episode reveals the role of chemistry in the multivariate assimilation: it acts as a strong constraint within 4D-Var, preventing it from assimilating erroneous observations.

### 4.6 Evaluation of non-observed species

Finally, the forecasts of two non-observed species issued from both data assimilation systems will be validated, $CH_4$ and $NO_x$,
the sum of $NO_2$ and NO, (Fig. 9). $CH_4$ CTM forecasts agree well with the MIPAS IMK/IAA data. And both data assimilation system keep generally this agreement, except the region around 2-1 hPa where 4D-Var develops an artificial bias related to the presence of $O_3$ model bias and the fact that $O_3$ data were assimilated in the upper stratosphere. As we saw this before, 4D-Var tends to develop such biases in many assimilated and non-assimilated species to compensate the $O_3$ bias. The problem of model $O_3$ bias is out of scope of the present article. We should only note that it can not be solved directly by data assimilation without an improved version of CTM (see Skachko et al, the paper in preparation to ACP for more details). $NO_x$ CTM forecasts are essentially different with data due to absent $NO_x$ sources in the model. As for $CH_4$, both data assimilation keep the model state unperturbed, except the region around 2-1 hPa where 4D-Var develops a bias for the same reason as stated above. Accidentally, this bias provides better agreement between the model and data in this region.

## 5 EnKF with cross correlations between species

All the EnKF experiments done so far used a brute force species localization, in other words, the sample covariance between species is set to zero. This type of localization should not be confused with the localization based on distance for the same species, which we keep. Now let us see what happens when we keep the sample cross-covariance intact.

To this end, we conducted an experiment where we assimilate $O_3$ and $N_2O$, two species that are not strongly related via the chemistry system. We will call this experiment the EnKF-CC, standing for EnKF with Cross-Covariances. In principle, we would expect that an observation of $O_3$ does not change significantly $N_2O$ and vice versa. In EnKF-CC, $O_3$ and $N_2O$ are put into a common observation space, defined by the observation vector $\mathbf{y}$ and the observation error covariance matrix $\mathbf{R}$. The ensemble of model vectors in observation space $H\mathbf{x}_i$ contain thus two blocks of $O_3$ and $N_2O$. This provides the cross-correlation terms in the model error covariance matrix $H\mathbf{B}H^T$ computed as

$$H\mathbf{B}H^T = \sum_i^N H(\mathbf{x}_i - \bar{\mathbf{x}})H(\mathbf{x}_i - \bar{\mathbf{x}})^T, \tag{10}$$

where $i \in [1, N]$ is the number of ensemble member, $N$ is the ensemble size, $\bar{\mathbf{x}}$ is the ensemble mean (see S14 for more details). The cross correlation terms between species remain after the localization of the error variance via the Schur product, because it filters out only the spurious spatial correlations.

Figure 10 shows an example of such EnKF-CC data assimilation comparing its results with EnKF discussed in the previous sections where the sample covariance between species was set to zero. The figure shows the $O_3$ OmF bias and standard deviation for EnKF (red) and EnKF-CC (yellow) analyses during 24h of September 15 2008. We observe that the EnKF-CC has noisy bias and an increased and noisy error standard deviations in the OmF correlations compared with the EnKF





experiment. A similar kind of impact is also obtained when we assimilate only one species and examine the OmF of the other non-observed species (results not shown). We thus conclude, as other studies have indicated, that the sample cross-covariance between weakly chemically related species, give rise to spurious analysis increments with a deterioration of the overall performance of the assimilation system.

**6   Conclusions**

We have conducted a comparison of an EnKF and 4DVar data assimilation system using a comprehensive stratospheric chemical transport model. Both data assimilation systems have online estimation of error variances based on the Desroziers' method and share the same correlation model for all prescribed error correlations (i.e. the background error for 4D-Var, initial error and model error for EnKF) so that each data assimilation system is nearly optimal and can also be compared to each other. A

previous comparison study by Skachko et al. (2014) showed that for chemical tracer transport both assimilation system provide results of essentially similar quality. This study examines in what way the inclusion of chemistry changes the performance of the assimilation system, but perhaps more importantly how an EnKF and a 4D-Var chemical data assimilations can be implemented in a real-life situation with several modelled and assimilated species. In this study we assimilate ozone, HCl, $HNO_3$, $H_2O$ and $N_2O$ observations from EOS Aura-MLS.

In the context of atmospheric chemistry, EnKF and 4D-Var differ in a number of ways. While 4D-Var, built on the assumption of a perfect model, tries to find a strong constraint solution that fits observations over a 24h window, EnKF on the other hand provides estimates at each model time step but allows for modelling error (mainly as a model error covariance). Furthermore, while 4D-Var infers information based on error correlation between observed and non-observed species, EnKF, on the other hand, introduces noise between weakly chemically-related species, and so far in practice, these cross-species error co-

variances are set to zero. So the question is: to which extent the chemical modelling is an important component of the analysis ? The implementation of a multi-species sequential chemical data assimilation is challenging by the need to properly tune and automate the estimation of a large number of input error parameters.

The comparison done in this paper shows that, in general, there is not a significant improvement in the OmF statistics of the system when the cross-correlation between species is kept (4D-Var) versus the EnKF system where the cross-species error

correlation has been filtered out. Differences do occur, however, when there is an important chemical modelling error or when there are large biases between model and observed values.

For example, the BASCOE CTM has an important model $O_3$ deficit near or above 1 hPa. The source of this model bias is unclear and is not discussed in this paper. The experiments show however that assimilating $O_3$ at these altitudes gives a poor agreement with observations. At these altitudes the chemical life-time of $O_3$ is smaller than the time step of the model and

consequently, any correction on the $O_3$ concentrations by the assimilation of $O_3$ measurements simply cannot correct for the model error. For the other species, such as HCl and $HNO_3$, the OmF statistics for EnKF are always better than for the 4D-Var. Two main reasons are responsible for this better performance. First, EnKF has a short-time forecast followed by frequent observational updates that is possibly more adequate for moderately fast chemical processes (but not for processes of life-time





smaller than the model time step). Second, the ensemble of CTM's provides better representation of the model variance. On the other hand, the cross-species covariances, implicit to a 4D-Var assimilation system, have a negative effect in the presence of strong model $O_3$ bias. The 4D-Var system tries to compensate the bias and thus develops small artificial biases in many chemically related with $O_3$ species, observed and non-observed. This is shown using OmF statistics for two observed species,

HCl and $H_2O$, and two non-observed, $CH_4$ and the $NO_x$ family.

The effect of large observation biases has a very different impact. For example, the EOS Aura MLS $N_2O$ has significant biases above 4 hPa. In this case, EnKF reaches the state close to observations from the first observation updates during the spin up phase, and keeps model close to observations afterwards because of short ensemble model forecasts and frequent observational updates. On the other hand, 4D-Var appears to be robust to erroneous observations. A significant number of

observations are rejected by the quality control, and in the end, 4D-Var provides analyses with more weight given to the model forecast rather than to the observations.

We have also examined the need to have cross-species localisation in an EnKF. Our study shows that the simultaneous assimilation of $O_3$ and $N_2O$, two species that are only weakly chemically related, gives rise to spurious cross-species error correlations that deteriorates the performance of EnKF, and it is then better to simply ignore those error correlations. To have

a more sensible approach to species localization could be the object of future work.

An important aspect of this study is the implementation of an online estimation of error variance parameters. The estimation of observation error variance and, in addition the background error variance for 4D-Var is done at each observation vertical level, using the Desroziers' method. The variance parameters being estimated are in fact very robust over time, showing little variability one day to the next.

Finally all the experiments were done with comparable wall clock time for EnKF and 4D-Var settings.

The study has also some limitations. An acknowledged difficulty often encountered in chemical data assimilation is the situation where both the model and the observations suffer from significant biases. This is the case for example with the BASCOE CTM CO and ClO when using the Aura MLS datasets. Solving this problem represents a challenging task that we have not conducted here, and would necessitate a dedicated study. Another limiting factor is the correlation length used in

this study. We have not attempted to estimate it, but rather have used what appears to be a reasonable value from past 4D-Var experiments. The estimated error variances and thus the weight given to the observations are also linked to the correctness of the error correlation, and this issue could also be investigated further.

## 7 Code availability

The numerical code of BASCOE CTM as well as two data assimilation methods, 4D-Var and EnKF, are provided upon an

email request to the authors.



*Author contributions.* S. Skachko and R. Ménard designed the experiments and S. Skachko carried them out. Q. Errera developed the codes of 4D-Var and the Desroziers' method. S. Skachko and Y. Christophe developed the EnKF code. S. Chabrillat and Y. Christophe worked on the CTM code. S. Skachko prepared the manuscript with contributions from all co-authors.

*Acknowledgements.* This research was financially supported at BIRA-IASB by the Belspo/ESA/PRODEX programme. S.Skachko acknowl-
5 edges fruitful discussions within the international "Study group on the added-value of chemical data assimilation in the stratosphere and upper-troposphere" at ISSI (International Space Science Institute) in Bern.



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





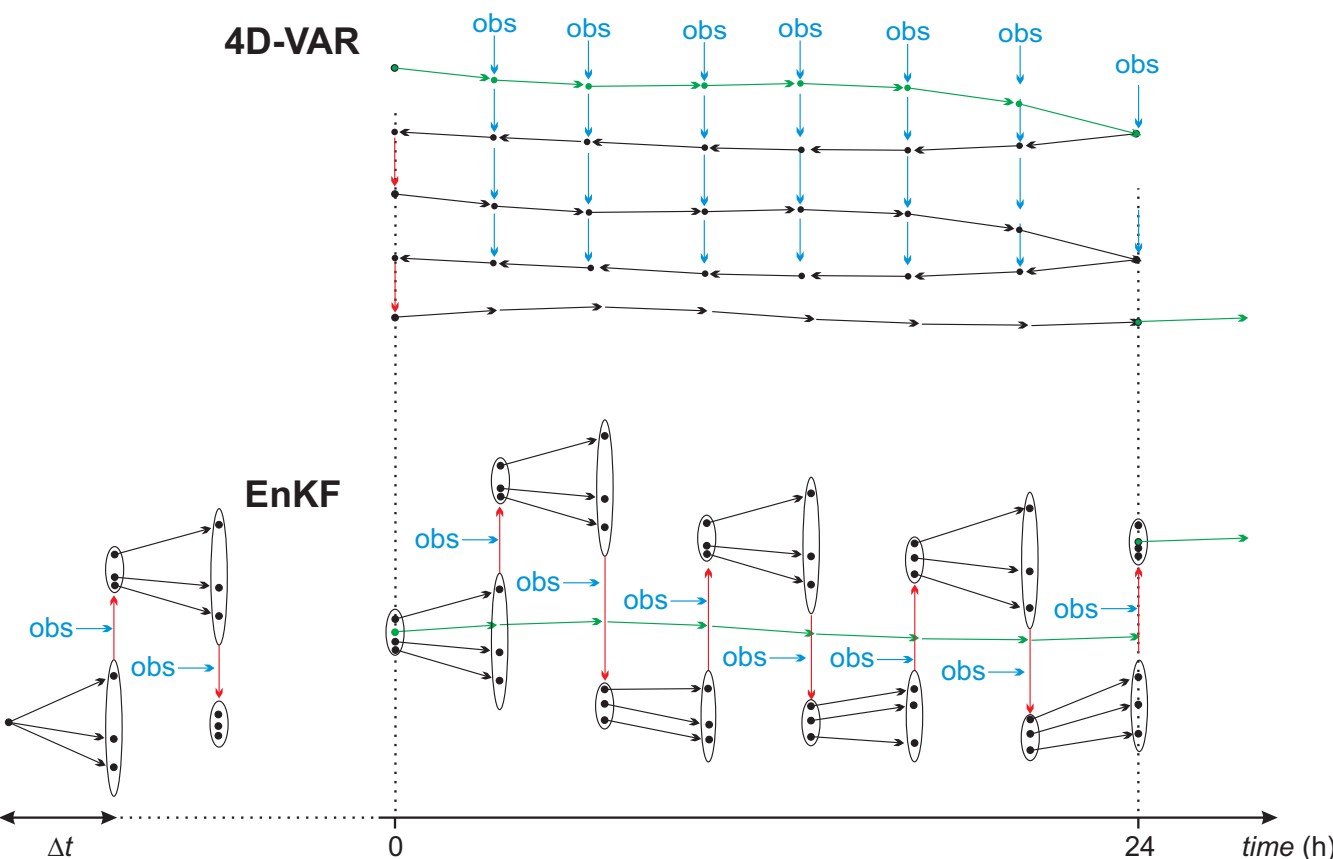

**Figure 1.** Schematic representation of the practical implementation of the 4D-VAR (top) and EnKF (bottom) assimilation methods in BAS-COE. Black dots represent model state and observational information is depicted in blue. The black arrows represent model integrations by one time step, vertical red arrows represent model state optimization (4D-VAR) or Kalman filter (EnKF). Green dots represent the analyses at 0h which are used as initial conditions for the diagnostic 24-h forecasts (green arrows). For clarity, the number of 4D-VAR iterations has been limited to 2 and the number of EnKF members has been limited to 3.





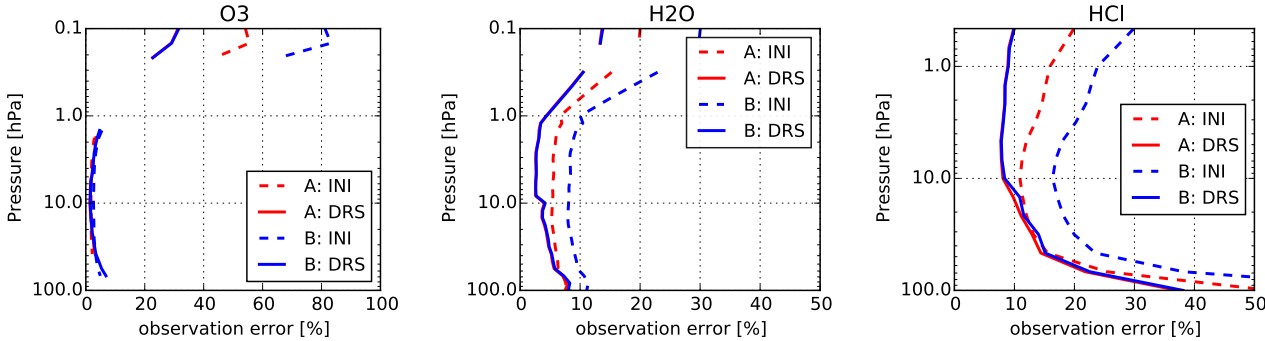

**Figure 2.** Initial (INI) observation error of experiment A (dashed red), starting with $\mathbf{R} = (\sigma_o)^2$, and B (dashed blue), starting with $\mathbf{R} = (1.5\,\sigma_o)^2$ and their Desroziers' (DRS) estimations (solid red and blue lines, respectively) using statistics of the first 24h. The statistics is shown for $O_3$, $H_2O$ and HCl.





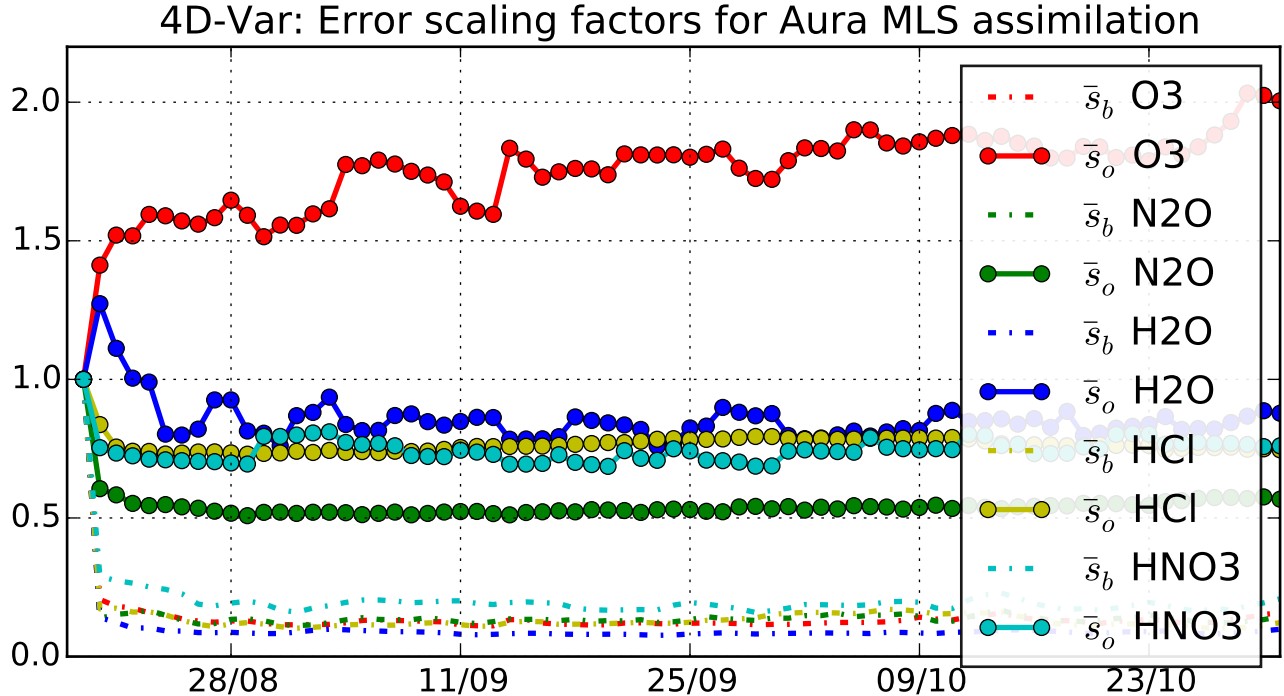

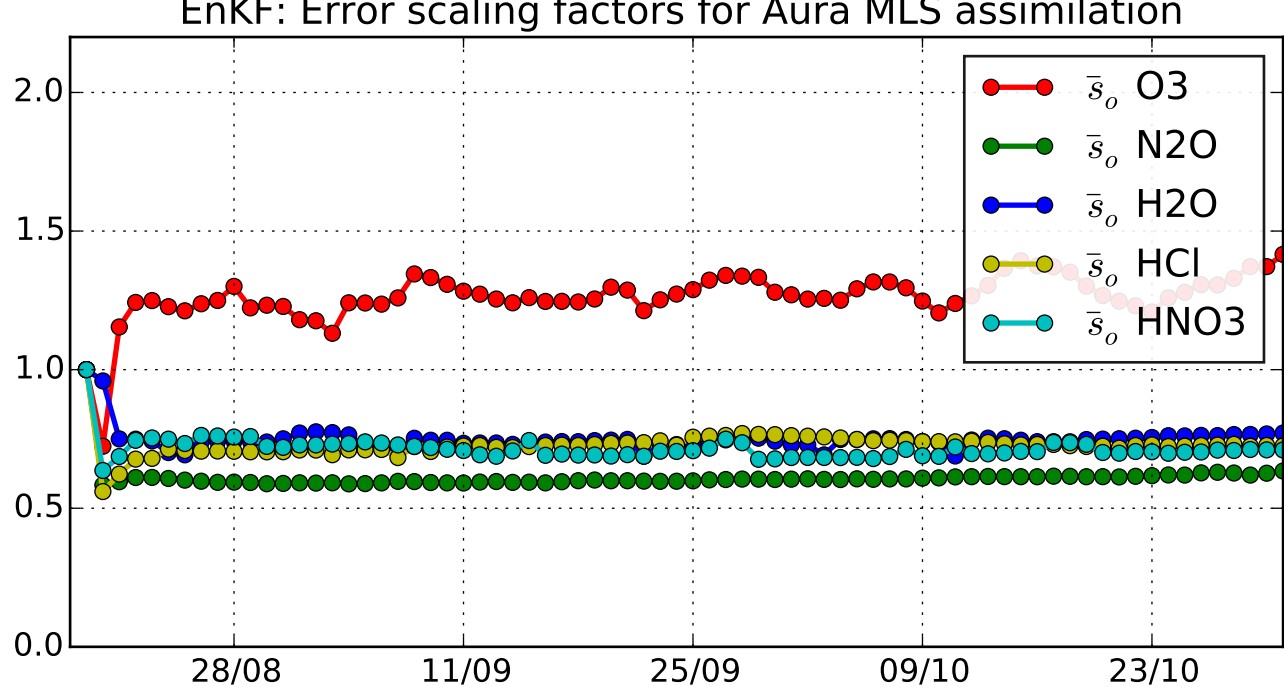

**Figure 3.** Estimated error scale factors within 4D-Var (left) and EnKF (right) for the period April-November 2008.





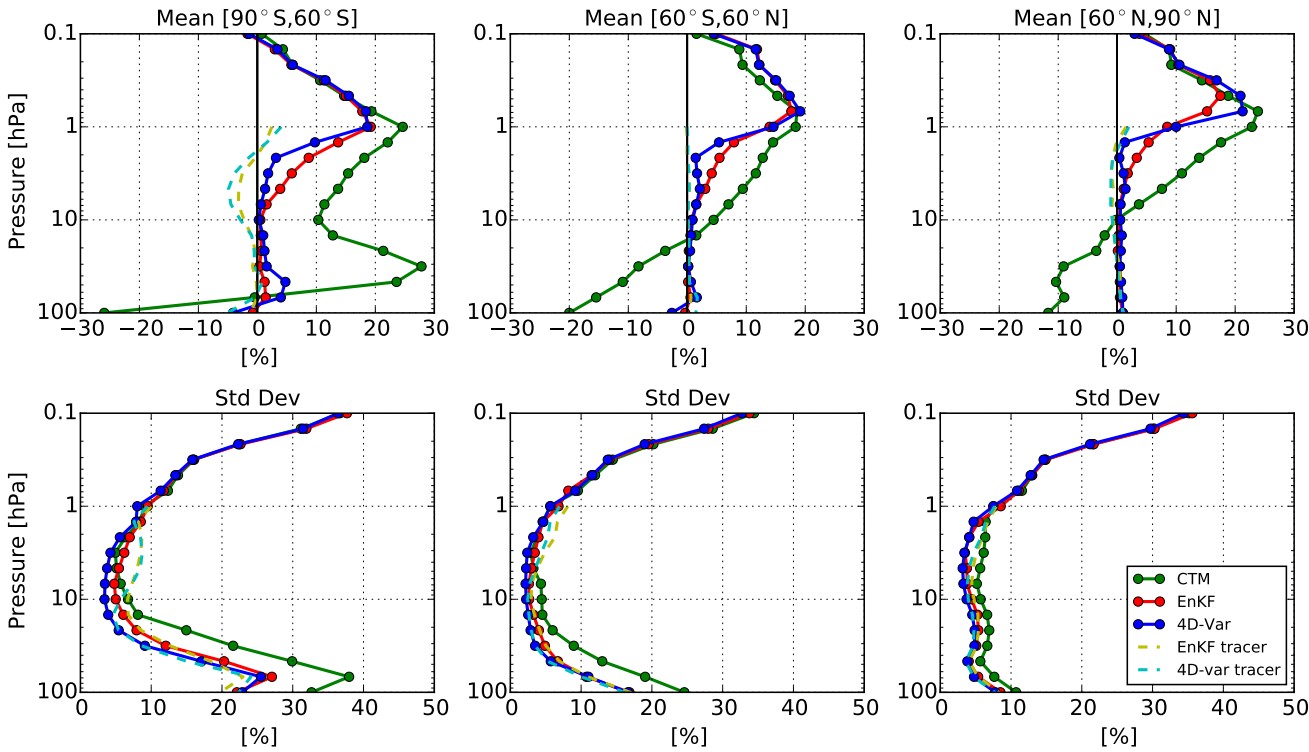

**Figure 4.** $O_3$ OmF bias (top) and standard deviation (bottom) computed for the full chemistry CTM (green), EnKF (red), 4D-Var (blue) based on the same model. The chemical tracer EnKF (dashed yellow) and 4D-Var (cyan) are also shown. OmF statistics is computed in percent with respect to the assimilated EOS Aura MLS data for the period September-October 2008 and for three latitude bands (from left to right: South Pole, Tropics - middle latitudes and North Pole).





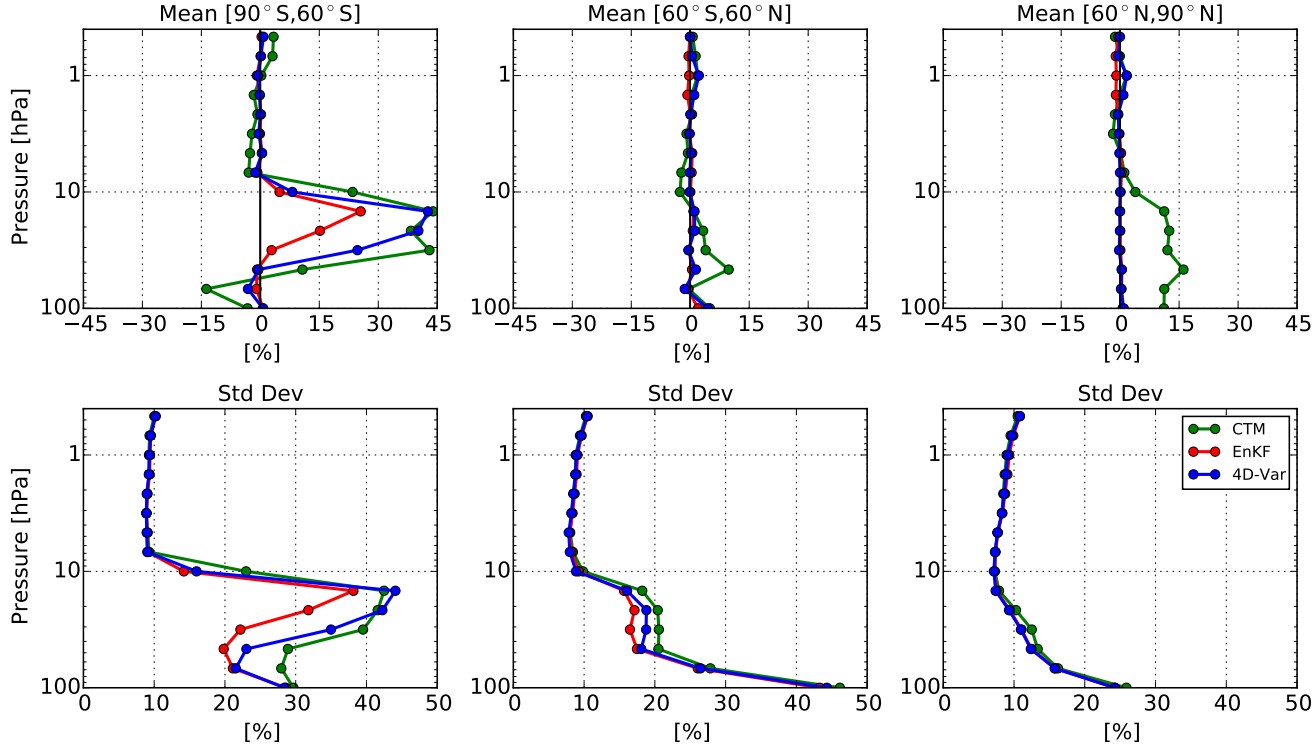

**Figure 5.** HCl OmF bias (top) and standard deviation (bottom) computed for the full chemistry CTM (green), EnKF (red) and 4D-Var (blue). OmF statistics is computed in percent with respect to the assimilated EOS Aura MLS data for the period May-June 2008 and for three latitude bands (from left to right: South Pole, Tropics - middle latitudes and North Pole).





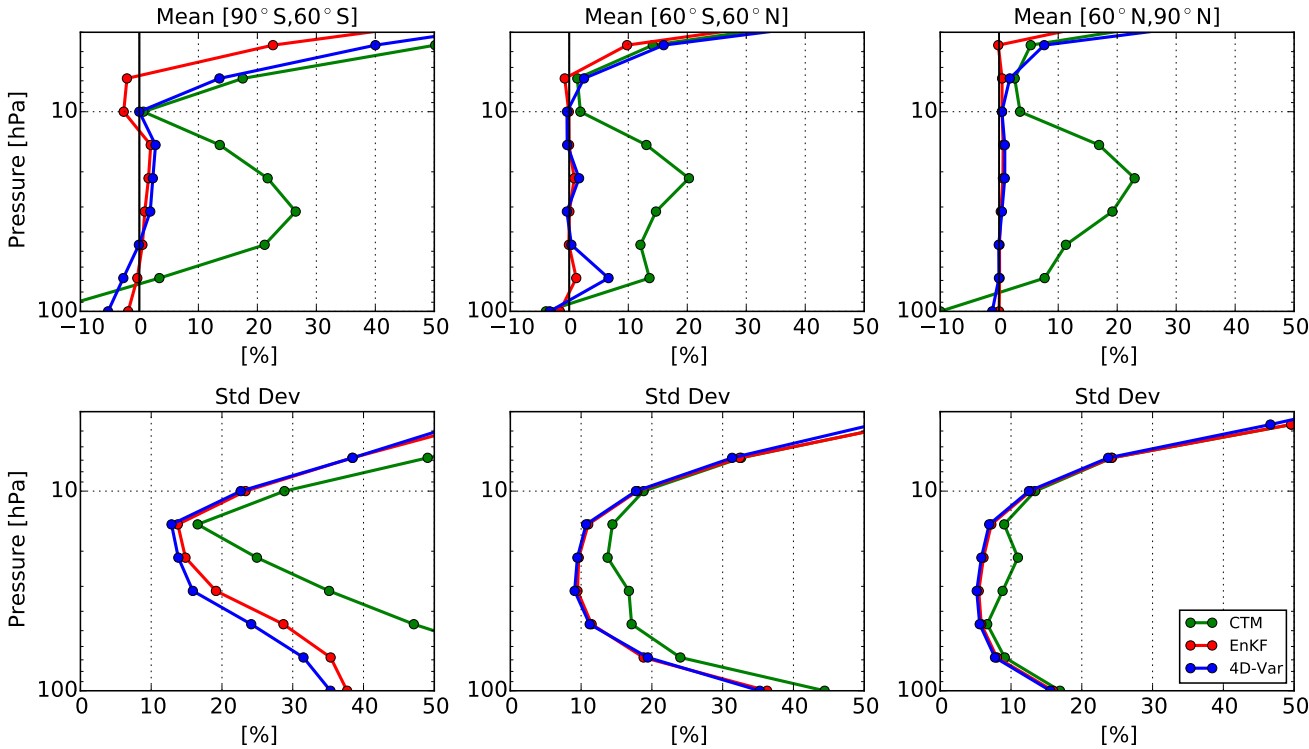

**Figure 6.** HNO$_3$ OmF bias (top) and standard deviation (bottom) computed for CTM (green), EnKF (red) and 4D-Var (blue). OmF statistics is computed in percent with respect to the assimilated EOS Aura MLS data for the period September-October 2008 and for three latitude bands (from left to right: South Pole, Tropics - middle latitudes and North Pole).





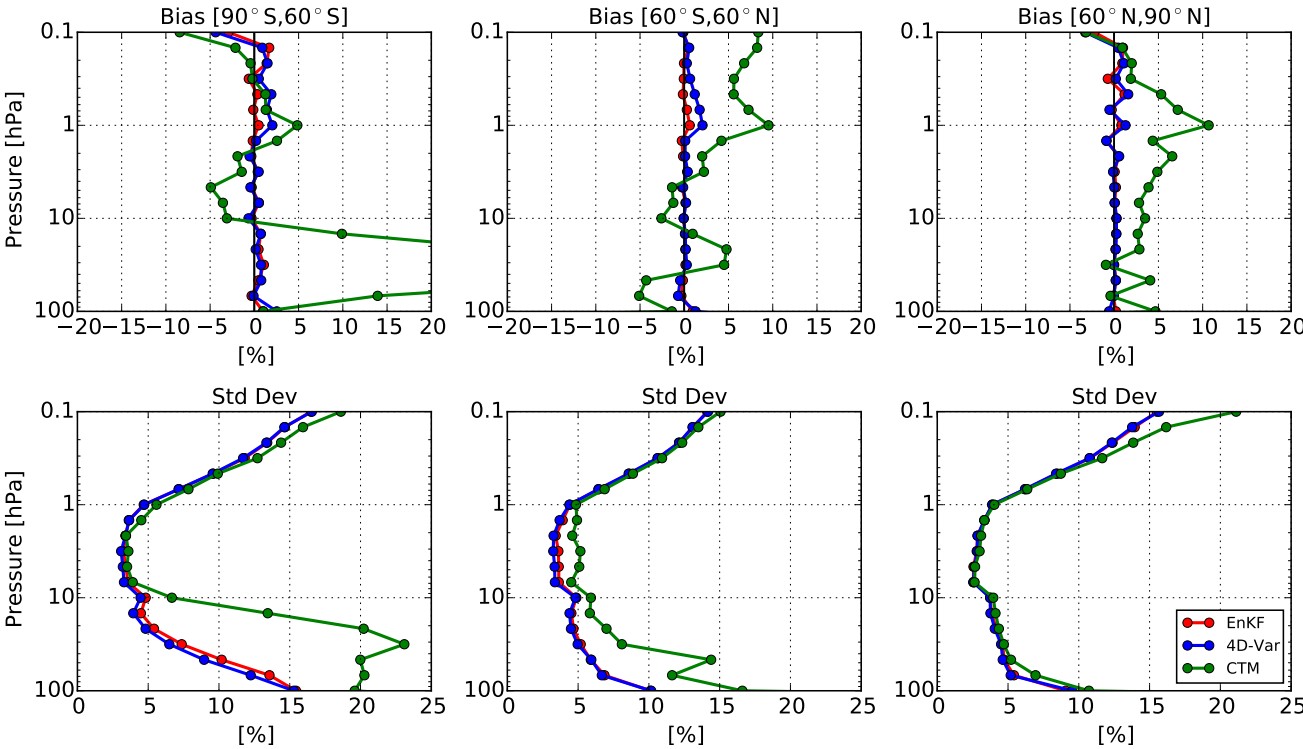

**Figure 7.** $H_2O$ OmF bias (top) and standard deviation (bottom) computed for the full chemistry CTM (green), EnKF (red) and 4D-Var (blue). OmF statistics is computed in percent with respect to the assimilated EOS Aura MLS data for the period September-October 2008 and for three latitude bands (from left to right: South Pole, Tropics - middle latitudes and North Pole).





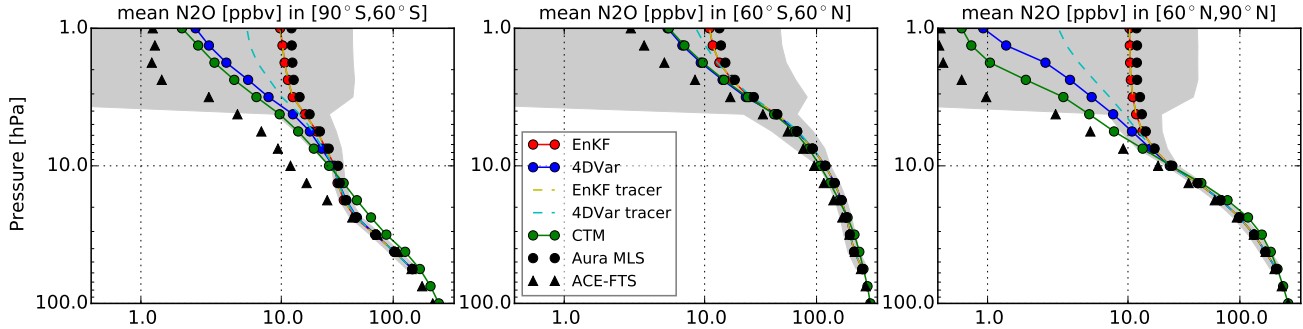

**Figure 8.** Mean $N_2O$ from the full chemistry CTM (green), 24 h forecasts from EnKF (red) and 4D-Var (blue), based on the same model, and Aura MLS (black dots) and ACE-FTS data (triangles). 24 h forecast from chemical tracer EnKF (dashed yellow) and 4D-Var (cyan) assimilation are also shown. The grey area shows the precision of Aura MLS data. The statistics are computed for September-October 2008 and for three latitude bands (from left to right: South Pole, Tropics - middle latitudes and North Pole).





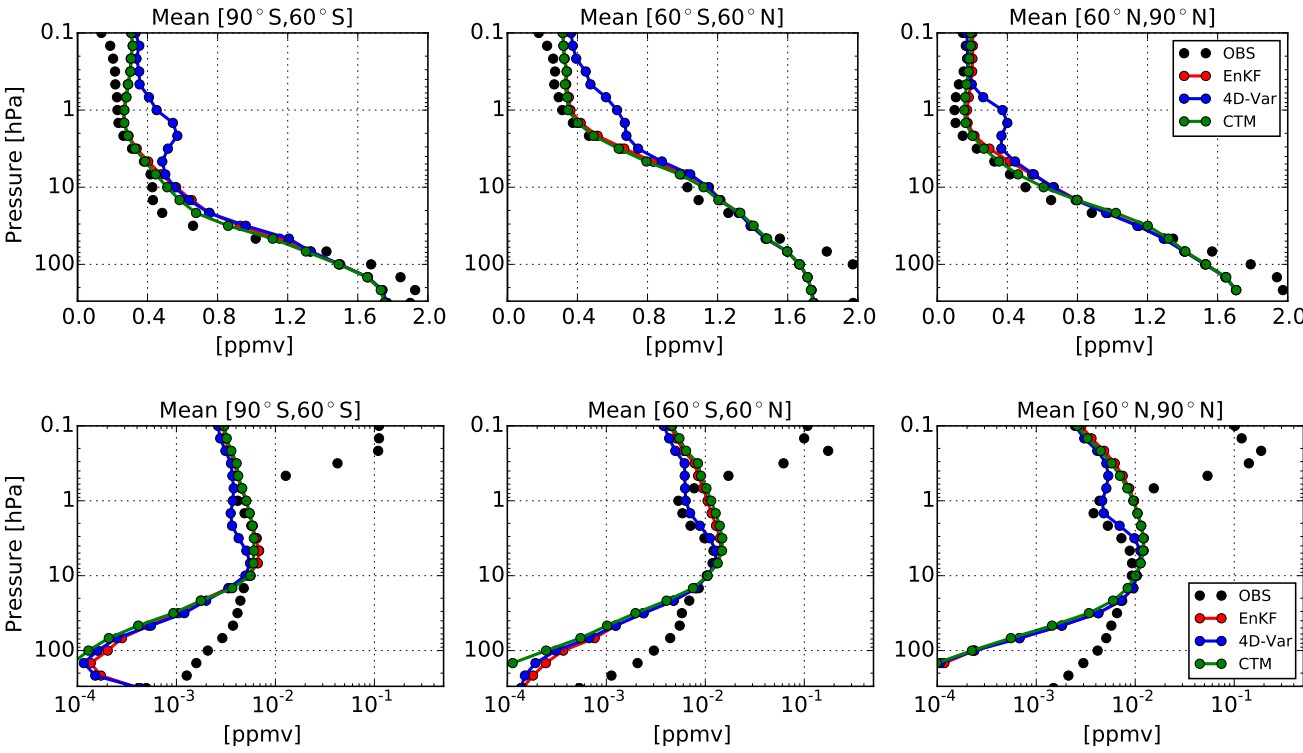

**Figure 9.** Verification of non-observed species from CTM (green), 24 h forecasts from EnKF (red) and 4D-Var (blue) assimilation against MIPAS IMK data (black dots): mean $CH_4$ (top) and mean $NO_x$ (bottom) profiles. The statistics are computed for September-October 2008 and for three latitude bands (from left to right: South Pole, Tropics - middle latitudes and North Pole).





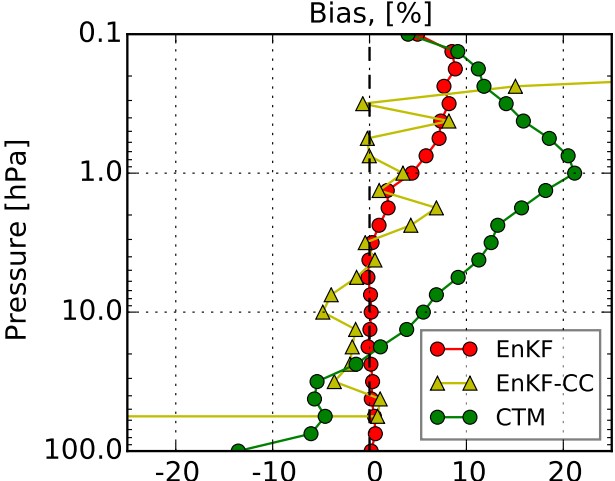

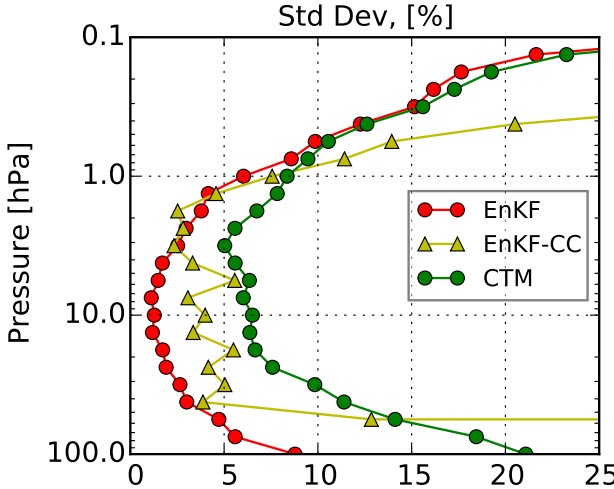

**Figure 10.** OmF bias (top) and standard deviation (bottom) between Aura MLS data and $O_3$ analyses of EnKF (red), EnKF-CC (yellow) and CTM (green), see text for acronym definition. The statistics are computed during 24h on September 15 2008.