# Peer review of "EnKF and 4D-Var Data Assimilation with the Chemistry Transport Model BASCOE (version 05.06)"

_Geoscientific Model Development, 2016_

## Referee Comment (RC1) · Anonymous Referee #1 · 13 May 2016

Review of paper: "EnKF and 4D-Var assimilation with a chemistry transport model"

by S. Skachko et al.

General

This paper represents a unique contribution on the comparison of EnKF and 4D-Var approaches for the assimilation of chemical species. It provides much insight, notably on issues related to inter-species error correlations and localization. I congratulate the authors. I provide here only minor suggested corrections.

Minor corrections:

L16: Change to: " one issue is the large number of …"

L19: Change "comparison reasons" to "comparison purposes"

L182: Change " will be a subject to" to "will be subject to"

L208: Define PSC

L211 Do not refer to MACC or define it

L303 Change "has the background quality" to "includes a background quality"

L 501 -504 Apparent contradiction where precision level 14 ppbv seems first to correspond to 38 % error at L 501 while at line 503 it corresponds to 250%. This is likely because in the first case it corresponds to 4.6 hPa level while in the other it corresponds to 1 hPa level. Sentence could be clearer.

L 698: why qualify as "tricky" the Aura-MLS dataset?

L707 End of sentence point needed after 5 hPa.

L709 " approaching to them", define what "them" represents. Not clear as is.

L727 Drop the reference to paper in preparation!

L733: Suggest to change "accidentally" by "incidentally"

L826 Life time of ozone is that small, lower than a model time step presumably of order 15 min?

Conclusion: Perhaps add as comment that an hybrid approach such as popular 4D-EnVar approach could emerge as a good way forward for chemical data assimilation.

---

## Referee Comment (RC2) · Anonymous Referee #2 · 17 May 2016

Review of EnKF and 4D-Var Data Assimilation with a Chemistry Transport Model

by S. Skachko et al.

This paper presents a comparison of the EnKF and 4D-Var data assimilation methods applied to a chemistry transport model. These two methods have been used extensively in applications to numerical weather prediction (NWP), a field with which I am more familiar. It is interesting to see such an intercomparison for an atmospheric chemistry model for simultaneously assimilating observations of the concentrations of several chemical species.

General Comments:

**1 Set up of assimilation windows**

One major concern I have is related to the basic setup of the experiments. Whereas intercomparisons in the context of NWP have generally used the same data assimilation window length for the two approaches (typically 6h), this study uses a different window length: 4D-Var is applied in its strong constraint formulation with a 24h window, whereas the EnKF is used to sequentially assimilate observations every 30 minutes (the time step length of the forecast model). In addition, model error perturbations are added every 30 minutes in the EnKF experiment, in contrast with the assumption of no model error over the 24h assimilation window in the 4D-Var setup. This difference in the configurations of the two methods seems to affect the conclusions of the study, as acknowledged by the authors near the end of the paper.

Alternative configurations could have possibly been chosen that would have reduced these differences. For example, the 4D-Var could have been implemented in its weak-constraint formulation, employing a model error term every 30 minutes based on the same covariance matrix as used for model error in the EnKF experiment. Alternatively, a 4D-EnKF approach could have been used with the EnKF in which ensembles of model solutions over a longer assimilation window are used to construct a 4D background error covariance matrix (as is done in most current implementations of both EnVar and the EnKF for NWP). In this case a window shorter than 24h (say 6h) would likely be preferable (and used for both EnKF and 4D-Var) to avoid problems with horizontal localization when a chemical species is advected within the time window by a distance comparable to the localization length scale. The authors should mention these two alternative approaches and discuss how they could have reduced the differences seen in some of their results.

As a consequence, the difference in window length also likely has a direct impact on the quality of the forecasts, since the 4D-Var forecasts are started from a state that may not be in as good agreement with the most recent observations as in the case of the EnKF. It would be useful to show the fit of the state used to initialize the 24h forecasts with respect to the observations

valid at the same time for the EnKF and 4D-Var experiments, or at least mention in the text how this fit differs between the two. I wonder if this explains some of the differences in results for HCl.

**2 Inter-species background error covariances**

The discussion regarding the estimation and treatment of inter-species background error covariances is somewhat unclear. The authors state that because it seems suitable to use the same model error covariances for each species it follows that the main source of error is from the transport. However, then the authors claim that the cross-species background error covariances are weak and not sufficiently well estimated with the EnKF ensemble and therefore should be set to zero when assimilating the observations. It seems to me that if the errors of the different species are dominantly affected by the same common source (i.e. the transport), then their errors should be correlated. Or maybe this is not the case due to the dependence of the concentration errors from transport on the concentration spatial gradients, which may be very different for each chemical species. Please provide some comment on this.

Also, I think the authors need to stress how the relatively small size of the ensemble (20 members versus the usual size for NWP of O(100)) could affect their conclusions regarding the utility of the cross-species covariances. Presumably with a much larger ensemble they would be better estimated and therefore could be more useful.

Specific comments:

In several places the term "observation errors" is used where I believe "observation error variances" or covariance or statistics is really what is meant. Similarly, on page 3, line 2 I am guessing that "the covariance inflation" refers to the inflation of the "background error covariance", please be more precise.

page 1, line 10 "…where we keep both estimates:" This is not clear. Is it meant that you keep both estimates fixed in time? Please clarify.

page 1, line 12 "…a single model error based…" It is difficult to decipher what is meant. I suppose this refers to using the same specified model error covariance matrix for each chemical species. Please make this more clear.

page 1, line 15 "…sampling noise errors…" and "These errors need to be filtered out…". This is awkward wording. Better to talk about "sampling error due to the use of a small ensemble leading to spurious covariances" and "setting these spurious covariance to zero".

page 1, line 20 "…not too small chemical life times…" This does not sound sufficiently quantitative. Could you instead say something like "chemical lifetimes longer than…" where you compare the lifetimes to some relevant time scale, e.g. the model time-step or assimilation window length.

page 2, line 6-7: It would make sense to also refer here to one of the first EnKF/4D-Var comparisons performed with real NWP systems done at the same center as one of the co-authors: Buehner et al. 2010 (2-part paper in MWR).

page 2, line 29-34: It is not clear why the issue of estimating error statistics for many chemical species is specific to the EnKF, as the text now seems to imply. Clearly this is equally important and challenging for the application of 4D-Var? Please clarify.

page 3, line 4 "background error" should be "background error covariance".

page 4, line 13, 15: Define the acronyms PSC and MACC.

page 5, line 15: It would help to state here that the same background error correlations are used for each chemical species and that the between-species covariances are assumed to be zero.

page 6, line 8: gamma is really set to 5? According to equation 5 this means an observation is rejected if its innovation is larger than 2.2 stddev (i.e. sqrt(5) = 2.2). Maybe equation 5 should have gamma^2 instead of just gamma?

page 6, line 20: the parameter N (ensemble size) needs to be defined here, since this is where it first appears.

page 6, line 23: "…are normally distributed random numbers…" should probably be "…are vectors of independent normally distributed random numbers…" otherwise equations 6 and 7 do not make sense.

page 6, lines 26-29: The two sentences about the application of the Desroziers method seems out of place here, since the method has not yet been introduced. Considering moving this to section 2.5.

page 7, lines 8-9: "To this end, the EnKF algorithm accounts now for a new effective procedure to find a current local sub domain in the model space." It is not at all clear what this sentence means. It must be better explained. Is the algorithm similar to that described by Houtekamer et al. (2014, MWR - "Parallel implementation of an EnKF")?

page 8, line 3: I believe "variance" should be "standard deviation". Please also check the entire paper to ensure standard deviation and variance are used correctly. Also, as already

mentioned, ensure the word "error" is not used in places where "error standard deviation" or "error statistics" or "error covariance" is actually what is meant. This is a common mistake that can be very confusing for some readers.

page 8, lines 32: "…and thus goes along in arguing that they represent some true error statistics." This does not sound very solid as a logical argument. Consider improving. Similarly for the following 2 sentences. It is probably not necessary to make this assertion at this point in the text.

page 9, line 4: "…10 iterations…" Is this enough to obtain a substantial reduction in the amplitude of the cost function gradient (i.e. at least a factor of 10)?

page 9, section 3: Please provide some additional general information about the observations assimilated: how much of the globe is observed during 24h? what is the horizontal and vertical spatial resolution?

page 10, section 4.1 and verification in general: What observations are used for verification? For example, are all observations with a valid time within 1h or 3h or ?h of the valid time of the forecast used? Since you are always verifying 24h forecasts valid at 0000UTC each day (if I understood correctly), does this tend to focus the verification in only certain geographical regions due to the orbit of the satellite?

page 11, line 16-17: related to the first general comment above, I think the difference seen here between the 4D-Var and EnKF results may be due to the difference in assimilation window lengths, please add a comment here.

page 12, line 32: "This is due to the automatic rejection by the 4D-Var of most observations…" What does this mean? Is it because the 4D-Var cannot make the forecast model solution fit the observations over the 24h assimilation window (due to model error) or is it referring to some quality control procedure (which I though was deactivated for this chemical species at the pressures considered here)?

page 13, line 23: "…model error covariance…" should be "…background error covariance in observation space…"

page 13, line 26: "…localization of the error variance…" should be "…localization of the error covariance…"

page 15, last paragraph: Another limitation is that much fewer ensemble members were used as compared with typical NWP applications. This should be mentioned.

---

## Author Comment (AC1) · 19 May 2016

[11pt,a4paper]article color

[Figure]

**Response to the Anonymous Referee 1**

May 19, 2016

The author's responses are marked in blue. We first would like to thank the anonymous referee 1 to the useful remarks.

General

This paper represents a unique contribution on the comparison of EnKF and 4D-Var approaches for the assimilation of chemical species. It provides much insight, notably on issues related to inter-species error correlations and localization. I congratulate the authors. I provide here only minor suggested corrections.

Minor corrections:

L16: Change to: "one issue is the large number of . . . "

Done.

L19: Change "comparison reason " to "comparison purposes "

Done.

L182: Change " will be a subject to " to "will be subject to "

Done.

L208: Define PSC

Done.

L211 Do not refer to MACC or define it

The MACC acronym has been defined.

L303 Change "has the background quality" to "includes a background quality "

Done.

L 501 -504 Apparent contradiction where precision level 14 ppbv seems first to correspond to 38% error at L 501 while at line 503 it corresponds to 250%. This is likely because in the first case it corresponds to 4.6 hPa level while in the other it corresponds to 1 hPa level. Sentence could be clearer.

We have modified the sentence as follows: "...the Aura MLS $N_2O$ precision is 24-14 ppbv (9-38%, relative to the observation mean at given altitude) and its accuracy is 70-3 ppbv (9-25%) in the pressure range 100-4.6 hPa but its precision drops to 14 ppbv (250%) at 1 hPa..."

L 698: why qualify as "tricky " the Aura-MLS dataset?

We replaced the word with "dubious Aura-MLS $N_2O$ dataset"

L707 End of sentence point needed after 5 hPa.

Done.

L709 " approaching to them", define what "them" represents. Not clear as is.

Replaced with "keeping the model closer to the assimilated observations"

L727 Drop the reference to paper in preparation!

Done.

[Figure]

L733: Suggest to change "accidentally " by "incidentally "

Done.

L826 Life time of ozone is that small, lower than a model time step presumably of order 15 min?

Indeed.

Conclusion: Perhaps add as comment that an hybrid approach such as popular 4D-EnVar approach could emerge as a good way forward for chemical data assimilation.

The following sentence is added to the last paragraph of the conclusions: "A future development of the BASCOE chemical data assimilation system would be a hybrid 4D-EnKF approach using the ensemble of models to construct a 4D background error covariance matrix."

---

## Author Comment (AC2) · 27 May 2016

[11pt,a4paper]article color

[Figure]

**Response to the Anonymous Referee 2**

May 27, 2016

The author's responses are marked in blue. We first would like to thank the anonymous referee 2 to the useful remarks.

This paper presents a comparison of the EnKF and 4D-Var data assimilation methods applied to a chemistry transport model. These two methods have been used extensively in applications to numerical weather prediction (NWP), a field with which I am more familiar. It is interesting to see such an intercomparison for an atmospheric chemistry model for simultaneously assimilating observations of the concentrations of several chemical species. General Comments:

1 Set up of assimilation windows

One major concern I have is related to the basic setup of the experiments. Whereas intercomparisons in the context of NWP have generally used the same data assimilation window length for the two approaches (typically 6h), this study uses a different window length: 4D-Var is applied in its strong constraint formulation with a 24h window, whereas the EnKF is used to sequentially assimilate observations every 30 minutes (the time step length of the forecast model). In addition, model error perturbations are added every 30 minutes in the EnKF experiment, in contrast with the assumption of no model error over the 24h assimilation window in the 4D-Var setup. This difference in

the configurations of the two methods seems to affect the conclusions of the study, as acknowledged by the authors near the end of the paper.

Alternative configurations could have possibly been chosen that would have reduced these differences. For example, the 4D-Var could have been implemented in its weak-constraint formulation, employing a model error term every 30 minutes based on the same covariance matrix as used for model error in the EnKF experiment. Alternatively, a 4D-EnKF approach could have been used with the EnKF in which ensembles of model solutions over a longer assimilation window are used to construct a 4D background error covariance matrix (as is done in most current implementations of both EnVar and the EnKF for NWP). In this case a window shorter than 24h (say 6h) would likely be preferable (and used for both EnKF and 4D-Var) to avoid problems with horizontal localization when a chemical species is advected within the time window by a distance comparable to the localization length scale. The authors should mention these two alternative approaches and discuss how they could have reduced the differences seen in some of their results.

As a consequence, the difference in window length also likely has a direct impact on the quality of the forecasts, since the 4D-Var forecasts are started from a state that may not be in as good agreement with the most recent observations as in the case of the EnKF. It would be useful to show the fit of the state used to initialize the 24h forecasts with respect to the observations valid at the same time for the EnKF and 4D-Var experiments, or at least mention in the text how this fit differs between the two. I wonder if this explains some of the differences in results for HCl.

We appreciate the concern raised by the referee to mention alternative 4D-EnVar approach (and it was done), however the purpose of this paper is not to try to develop a hybrid method, but first to compare two data assimilation approaches in the set-up that is usually found in chemical applications. It is current practice to use a 24 h window in stratospheric 4D-Var chemistry because of diurnal cycles with many chemical species and because of the polar orbiter limb sounder whose spatial coverage is far too limited

in 6h, whereas a 24 h does provide an acceptable complete spatial coverage. The BASCOE 4D-Var system has been running in near-real time since the early 2000's using the 24 h assimilation window. On the other hand, since the early 2000's sequential methods, such as simplified KF and later with the EnKF, have been running with model error and with a time window of 1 hour. In our previous work, we did compare 4D-Var and EnKF but for chemical transport with no chemistry (Skachko et al 2014), referred as S14, and found nearly identical results, despite the fact that BASCOE EnKF used 0.5 h ensemble model forecasts and 4D-Var, the 24 h assimilation window. From a dynamical system point of view, chemical transport and NWP are quite different. We do not find growth of error in chemical transport. And adding chemistry produces a system with a rather dissipative behaviour. This is corroborated by the fact that chemical transport model simulation (without chemical data assimilation) shows stable differences with chemical observations over periods of months and even years. So we disagree that the difference in window length has such an impact in the context of chemical transport. What we did here is to add chemistry to this problem and see what are the implications due to chemistry. In particular we acknowledge that the window length and having model error or not does have an impact on certain species. However, in the context where the chemistry would be directly coupled with an NWP model, a 6hr time window would need to be considred. The cost of operational implementation of such an approach would be nevertheless prohibitive. Lower resolution, one way coupling or simplified chemistry (which displays very different characteristics in terms of model error) are most likely to be considered for such set-up.

2 Inter-species background error covariances The discussion regarding the estimation and treatment of inter-species background error covariances is somewhat unclear. The authors state that because it seems suitable to use the same model error covariances for each species it follows that the main source of error is from the transport. However, then the authors claim that the cross-species background error covariances are weak and not sufficiently well estimated with the EnKF ensemble and therefore should be set to zero when assimilating the observations. It seems to me that if the errors of

the different species are dominantly affected by the same common source (i.e. the transport), then their errors should be correlated. Or maybe this is not the case due to the dependence of the concentration errors from transport on the concentration spatial gradients, which may be very different for each chemical species. Please provide some comment on this. Also, I think the authors need to stress how the relatively small size of the ensemble (20 members versus the usual size for NWP of O(100)) could affect their conclusions regarding the utility of the cross-species covariances. Presumably with a much larger ensemble they would be better estimated and therefore could be more useful.

Although it is the same transport that acts on all species, the resulting distribution of species are not necessarily correlated. Long-lived species for instance which are the best candidates for such correlations show in some cases complicated correlations patterns that depend on latitude and height and vary in time (e.g. Sankey and Shepherd 2003). Besides, preliminary experiments with BASCOE-4D-Var system showed that the cross-covariance between innovations of long-lived species is rather noisy and assimilation experiments that accounts for cross-covariance between long-lived species in the background error term do not show in practice significant improvements in the analysis quality. This additional information is added in the article text.

An example of weakly related chemical species is considered in the article: the ozone and $N_2O$ are assimilated taking into account the cross-correlation terms in the background error covariance matrix. The spurious errors, that arise from the use of a rather small ensemble, can be reduced by using larger ensemble, however the NWP ensemble of O(100) would not be sufficient to solve the problem. And the computation cost of such ensemble data assimilation becomes unreasonably expensive. To our knowledge, all groups working on chemical data assimilation simply put the cross-correlations between weakly chemically related species to zero. We don't try to solve this problem, but only provide an illustration. The solving of the automatic localization procedure between species would be a subject of another study which is out of scope of the current

paper.

Specific comments:

In several places the term "observation errors" is used where I believe "observation error variances" or covariance or statistics is really what is meant.

Done.

Similarly, on page 3, line 2 I am guessing that "the covariance inflation" refers to the inflation of the "background error covariance", please be more precise.

The sentence is rewritten as: "In the same line of thought, the Desroziers' method (Desroziers, 2005) was also used to simultaneously estimate the covariance inflation factor and the observation error variance(Li et al. 2009, Gaubert et al. 2014)."

page 1, line 10 "Æůhere we keep both estimates:" This is not clear. Is it meant that you keep both estimates fixed in time? Please clarify.

We wanted to say that in the 4D-Var experiments, both background and observation error covariance matrices were estimated using the Desroziers' method. To be more clear on this, I reformulated the sentence as follows: "For comparison purposes, we apply the same estimate procedure in the 4D-Var data assimilation, where both, the background and observation error covariance matrices are estimated using the Desroziers' method."

page 1, line 12 "Æą single model error basedÅğ' It is difficult to decipher what is meant. I suppose this refers to using the same specified model error covariance matrix for each chemical species. Please make this more clear.

This part is reformulated as: "However in EnKF, the background error covariance is modelled using the full chemistry model and a model error term which is tuned using an adjustable parameter. We found that it is adequate to have the same value of this parameter based on the chemical tracer formulation that is applied for all observed

species."

page 1, line 15 "Æşampling noise errorsÅğ' and "These errors need to be filtered outÅğ'. This is awkward wording. Better to talk about "sampling error due to the use of a small ensemble leading to spurious covariances" and "setting these spurious covariance to zero".

This is rewritten in one sentence as: "The second issue in EnKF with comprehensive atmospheric chemistry models is the spurious error, that occurs when species are weakly chemically related at the same location."

page 1, line 20 "Æőot too small chemical life timesÅğ' This does not sound sufficiently quantitative. Could you instead say something like "chemical lifetimes longer thanÅğ' where you compare the lifetimes to some relevant time scale, e.g. the model time-step or assimilation window length.

This is rewritten as: "If the erroneous chemical modelling is associated with moderately fast chemical processes, but whose life-times are longer than the model time step, then EnKF performs better,..."

page 2, line 6-7: It would make sense to also refer here to one of the first EnKF/4D-Var comparisons performed with real NWP systems done at the same center as one of the coauthors: Buehner et al. 2010 (2-part paper in MWR).

Done.

page 2, line 29-34: It is not clear why the issue of estimating error statistics for many chemical species is specific to the EnKF, as the text now seems to imply. Clearly this is equally important and challenging for the application of 4D-Var? Please clarify.

Our previous work, S14, based on a chemical tracer version of the CTM showed that the EnKF is much more sensitive to the parametrization of the error statistics than the 4D-Var. In order to make it more clear, I put the following text at the end of the next paragraph: "Contrary to the EnKF, the 4D-Var is much more tolerant to the parametriza-

tion of the error statistics, as it was shown in S14. Hence, the online estimation of the error statistics is of great importance only for EnKF."

page 3, line 4 "background error" should be "background error covariance".

Done.

page 4, line 13, 15: Define the acronyms PSC and MACC.

Done.

page 5, line 15: It would help to state here that the same background error correlations are used for each chemical species and that the between-species covariances are assumed to be zero.

The sentence of line 24-25: "...$\mathbf{\Lambda}^{1/2}$ is the spatial correlation matrix defined on a spherical harmonic basis hence diagonal;" is modified as follows: "...$\mathbf{\Lambda}^{1/2}$ is the spatial correlation matrix, identical for each chemical species, defined on a spherical harmonic basis hence diagonal;" And a new sentence is added at the end of the same paragraph: "The between-species covariances are assumed to be zero in the background error covariance matrix $\mathbf{B}_0$."

page 6, line 8: gamma is really set to 5? According to equation 5 this means an observation is rejected if its innovation is larger than 2.2 stddev (i.e. sqrt(5) = 2.2). Maybe equation 5 should have $\gamma^2$ instead of just $\gamma$?

Yes, $\gamma^2$ is written now in the equation.

page 6, line 20: the parameter N (ensemble size) needs to be defined here, since this is where it first appears.

The parameter N is now defined here.

page 6, line 23: "Æąre normally distributed random numbersÅğ' should probably be "Æąre vectors of independent normally distributed random numbersÅğ' otherwise

equations 6 and 7 do not make sense.

Done.

page 6, lines 26-29: The two sentences about the application of the Desroziers method seems out of place here, since the method has not yet been introduced. Considering moving this to section 2.5.

The text is modified: "...where the adjustable scaling factor $s_o(i)$ is estimated using the method described in Sect.2.5". And the second sentence is written as: "Besides, the current version of EnKF allows for more accurate observation error covariance estimation with respect to S14 because it computes $s_o(i)$ as a function of observation vertical pressure level."

page 7, lines 8-9: "To this end, the EnKF algorithm accounts now for a new effective procedure to find a current local sub domain in the model space." It is not at all clear what this sentence means. It must be better explained. Is the algorithm similar to that described by Houtekamer et al. (2014, MWR - "Parallel implementation of an EnKF")?

The sentence is now written as: "To this end, the EnKF algorithm accounts now for a procedure to find a current local sub domain in the model space using the K-D tree, which is a binary search tree where the comparison key is cycled between K components (K = 3 in our case, because the observation location is a 3-dimensional vector). More information on the method can be found on the Web or in any textbook on data structures (e.g. Gonnet and Baeza-Yates 1991)."

page 8, line 3: I believe "variance" should be "standard deviation". Please also check the entire paper to ensure standard deviation and variance are used correctly. Also, as already mentioned, ensure the word "error" is not used in places where "error standard deviation" or "error statistics" or "error covariance" is actually what is meant. This is a common mistake that can be very confusing for some readers.

Done.

Interactive
comment

page 8, lines 32: "Æand thus goes along in arguing that they represent some true error statistics." This does not sound very solid as a logical argument. Consider improving. Similarly for the following 2 sentences. It is probably not necessary to make this assertion at this point in the text.

The text is modified as: "The time evolution of the error variance scaling factors at individual levels is, as in Figure 3, (result not shown) generally consistent over time, especially for the EnKF estimates. Furthermore, we note that for the EnKF results, the scaling factors of any species show no drift in time. We argue from this result that there is apparently no need to have a different model error $\alpha$ for different species. Thus we conclude that for a chemical transport models, the main source of model error can be attributed to transport errors primarily."

page 9, line 4: "Åś0 iterationsÅğ' Is this enough to obtain a substantial reduction in the amplitude of the cost function gradient (i.e. at least a factor of 10)?

The reduction in the gradient of the cost function depends on the assimilated observations. In the case of EOS Aura-MLS data, the reduction varies between the factor of 6 to 10. The reduction is more important when the number of iteration is doubled, however it does not lead to any significant improvement in the OmF statistics. For the practical purposes, the near-operational BASCOE 4D-Var system used in the MACC project is run with 10 iterations, because the chemical modelling is essentially time-consuming task. In a similar data assimilation case, Elbern et al. (2010) utilize between 12 and 16 iterations.

page 9, section 3: Please provide some additional general information about the observations assimilated: how much of the globe is observed during 24h? what is the horizontal and vertical spatial resolution?

The 24h of the 4D-Var assimilation window is chosen because within this period, Aura-MLS data provide near-global coverage with gaps that are dispersed regularly within the correlation length of the data assimilation, i.e. 800km. The following text is added:

"The EOS Aura-MLS data cover the latitude range between 82°S and 82°N with an along-track separation of around 165 km between consecutive scans. Around 3500 vertical scans are performed every day. The vertical resolution varies for different species."

page 10, section 4.1 and verification in general: What observations are used for verification? For example, are all observations with a valid time within 1h or 3h or ?h of the valid time of the forecast used? Since you are always verifying 24h forecasts valid at 0000UTC each day (if I understood correctly), does this tend to focus the verification in only certain geographical regions due to the orbit of the satellite?

At every 0.5 h model time step of the 24 h forecast, the BASCOE system computes the OmF statistics. Thus, figures 4-10 show the statistics where all observations between 0 h and 24 h UTC, distributed within a given latitude band and chosen period, are taken into account.

page 11, line 16-17: related to the first general comment above, I think the difference seen here between the 4D-Var and EnKF results may be due to the difference in assimilation window lengths, please add a comment here.

In the context of chemistry, the difference in data assimilation window lengths really has implications, as pointed out by the referee. Our conclusions, page 14, line 32-32 and page 15 line 1: "Two main reasons are responsible for this better performance. First, EnKF has a short-time forecast followed by frequent observational updates that is possibly more adequate for moderately fast chemical processes (but not for processes of life-time smaller than the model time step). Second, the ensemble of CTM's provides better representation of the model variance."

page 12, line 32: "This is due to the automatic rejection by the 4D-Var of most observations.." What does this mean? Is it because the 4D-Var cannot make the forecast model solution fit the observations over the 24h assimilation window (due to model error) or is it referring to some quality control procedure (which I though was deactivated

for this chemical species at the pressures considered here)?

Yes, the background quality check (BgQC) procedure rejects the observations. BgQC is active for both EnKF and 4D-Var data assimilation systems, though it works differently for each systems when their background states differ.

page 13, line 23: "..model error covariance.." should be "..background error covariance in observation space.."

Done.

page 13, line 26: "..localization of the error variance.." should be "..localization of the error covariance.."

Done.

page 15, last paragraph: Another limitation is that much fewer ensemble members were used as compared with typical NWP applications. This should be mentioned.

The ensemble used in this study is typical for chemical data assimilation. Here, we don't aim to compare BASCOE system with NWP applications.

**References**

Elbern, H., J. Schwinger, and R. Botchorishvili (2010), Chemical state estimation for the middle atmosphere by four‐dimensional variational data assimilation: System configuration, J. Geophys. Res., 115, D06302, doi:10.1029/2009JD011953.

Skachko, S., Errera, Q., Ménard, R., Christophe, Y., and Chabrillat, S.: Comparison of the ensemble Kalman filter and 4D-Var assimilation methods using a stratospheric tracer transport model, Geosci. Model Dev., 7, 1451-1465, doi:10.5194/gmd-7-1451-2014, 2014.

Correlations of long-lived chemical species in a middle atmosphere general circulation model, J. Geophys. Res., 108, 4494, doi:10.1029/2002JD002799, D16.

---

## Referee Comment (RC3) · Anonymous Referee #2 · 30 May 2016

Follow-up review of "EnKF and 4D-Var Data Assimilation with a Chemistry Transport Model" by S. Skachko et al.

General comments:

I am confused by the authors' response to my main concern, related to the difference in the window length used for the 4D-Var and EnKF experiments. In response to my first general comment, the authors' response is:

"So we disagree that the difference in window length has such an impact in the context of chemical transport."

Then, when I later bring up the same point again in relation to the discussion of the results, the authors' response is:

"In the context of chemistry, the difference in data assimilation window lengths really has implications, as pointed out by the referee."

Also, I believe the authors' misinterpreted part of my first general comment. I made no suggestion that a hybrid 4D-EnVar experiment be performed, or even mentioned. What I did suggest was that a 4D-EnKF approach (with model error perturbations only applied at the beginning of each window to be equivalent with strong-constraint 4D-Var) should be considered and mentioned, since this would allow a longer window to be used for the EnKF. In this case, the analysis would be forced to simultaneously fit all of the observations distributed over a longer window, while still satisfying the model equations, as in 4D-Var.

I appreciate that the authors have tested two data assimilation methods in configurations as they are usually used for chemical applications. This point should be emphasized in the paper to justify the choice. However, it would be helpful to inform the reader that other configurations are possible that would reduce the differences between the two approaches (i.e. including model error in weak-constraint 4D-Var and using 4D covariances with a longer window in the EnKF). Otherwise, readers will conclude that one approach (i.e. EnKF or 4D-Var) is fundamentally better or worse than the other in some respects, whereas it is more likely the choice of how each approach was implemented that is more important.

Specific comments:

In response to the third specific comment, your revised sentence seems imprecise: "For comparison purposes, we apply the same estimate procedure in the 4D-Var data assimilation, where both, the background and observation error covariance matrices are estimated using the Desroziers' method." I presume it is only the scale factors for both covariance matrices that are estimated and not the full matrices? Please improve the wording.

In response to the fifth specific comment, your revised sentence does not clear up my concern: "The second issue in EnKF with comprehensive atmospheric chemistry models is the spurious error, that

occurs when species are weakly chemically related at the same location." The term "spurious error" is very ambiguous... how can error be spurious? I believe this is again where "error" is used in place of "error covariance". Only the "estimated error covariance" is spurious. [The word "error" on its own really should be reserved for the difference between an estimate and the truth and I don't think this is what is meant in this case. I realize that some published papers have used "error" to mean "error standard deviation" or "error covariance", but I believe this has needlessly caused confusion for some people in the DA community.]

---

## Referee Comment (RC4) · Anonymous Referee #2 · 3 Jun 2016

I am satisfied with the authors responses. I only suggest the following text be used instead of the added text on page C4 of the authors latest response (which incorrectly referred to the "4D-Var" window when describing the 4DEnKF approach):

"Other possibilities may be considered to properly compare two essentially different data assimilation systems. For example, the 4DEnKF (Hunt et al., 2004) approach could be used that computes 4D error covariances from the ensemble of forecasts at several times within the assimilation window. This would allow a longer assimilation window to be used in the EnKF experiment, making it more comparable to the configuration of 4D-Var. In this case, the EnKF analysis would be forced to simultaneously fit all of the observations distributed over a longer window, while still satisfying the model equations, as in 4D-Var. Applying the model error in this 4DEnKF only at the beginning of each assimilation window would make it similar to the strong-constraint version of

4D-Var that was used."

Hunt, B. R., and Coauthors, 2004: Four-dimensional ensemble Kalman filtering. Tellus, 56A, 273–277.

———————————————

---

## Referee Comment (RC5) · Anonymous Referee #3 · 3 Jun 2016

Review of EnKF and 4D-Var Data Assimilation with a Chemistry Transport Model by S. Skachko et al.

This study compares EnKF and 4D-Var data assimilation methods applied to a chemistry transport model. The purpose is to compare relative merits of the two methods on long time (short windows) atmospheric chemistry data assimilation with prescribed flow fields.

Major comments:

1 EnKF Experimental setup: Page 6: "the model error term is added to observed species only." What is the rationale for this? The same L operator seems to be used both for 4D-Var and EnKF, but at lest in the definition of \eta in (7), and (1) or (2), L lives in different spaces.

[Figure]

2 The authors claim that the same error covariances are used in both cases [page 14: "the same correlation model for all prescribed error correlations (i.e. the background error for 4D-Var, initial error and model error for EnKF)"]; however, on page 8 around line 10, they seem to indicate different localization operators that come in to build B. This should be clarified.

3 Cross species localization: In Section 5 the authors discuss the effects of inter-species localization. It is unclear to me what is done here. Is ENFK-CC the same as EnKF except that in EnKF-CC the O3 and NO2 are localized? If that is the case, then this is problematic because one cannot choose to localize some species and not localize the others because it introduces transients that may lead to spurious bias oscillations. This should be clarified as well.

Minor comment:

Page 4 line 2: "cross-covariance between species are taken into account automatically using the 4D-Var adjoint mode" is not clear to me. How is this achieved?
* * *

---

## Author Comment (AC3) · 3 Jun 2016

**Response to the Follow-up review of the Anonymous Referee 2**

June 3, 2016
We thank the referee for the detailed revision of the paper. The author's responses are marked in blue.

General comments:

I am confused by the authors' response to my main concern, related to the difference in the window length used for the 4D-Var and EnKF experiments. In response to my first general comment, the authors' response is: "So we disagree that the difference in window length has such an impact in the context of chemical transport." Then, when I later bring up the same point again in relation to the discussion of the results, the authors' response is: "In the context of chemistry, the difference in data assimilation window lengths really has implications, as pointed out by the referee."

We should be more clear on this. The first mentioned sentence means that in the context of chemical tracer transport only (without chemistry system), there is no difference in using an EnKF with 30 min ensemble model forecasts and a model error term or a 4D-Var with 24 h assimilation window without model error term. This was shown in our previous article (Skachko et al 2014). The purpose of the present work is to reveal

the role of the chemistry system (including interactions between chemical species) in the context of our two data assimilations that are configured as they are normally used in chemical data assimilation applications: one model time step ensemble model forecasts within EnKF, and 12 - 24 h of 4D-Var assimilation window.

Also, I believe the authors' misinterpreted part of my first general comment. I made no suggestion that a hybrid 4D-EnVar experiment be performed, or even mentioned. What I did suggest was that a 4D-EnKF approach (with model error perturbations only applied at the beginning of each window to be equivalent with strong-constraint 4D-Var) should be considered and mentioned, since this would allow a longer window to be used for the EnKF. In this case, the analysis would be forced to simultaneously fit all of the observations distributed over a longer window, while still satisfying the model equations, as in 4D-Var.

I appreciate that the authors have tested two data assimilation methods in configurations as they are usually used for chemical applications. This point should be emphasized in the paper to justify the choice. However, it would be helpful to inform the reader that other configurations are possible that would reduce the differences between the two approaches (i.e. including model error in weakconstraint 4D-Var and using 4D covariances with a longer window in the EnKF). Otherwise, readers will conclude that one approach (i.e. EnKF or 4D-Var) is fundamentally better or worse than the other in some respects, whereas it is more likely the choice of how each approach was implemented that is more important.

The fourth paragraph of the introduction is modified as follows: "But how do the EnKF and the 4DVar methods compare when photochemical reactions are taken into account? Do the results depend on the assimilated chemical species? Using actual satellite datasets and operational configurations, what are their respective performances in terms of precision, accuracy and computational efficiency? What is the role of the practical implementation of each method, when the full description of the stratospheric chemistry is taken into account in the CTM. These are the main questions addressed

in this paper."

The conclusions start with: "We have conducted a comparison of an EnKF and 4DVar data assimilation system using a comprehensive stratospheric chemical transport model. We considered 4D-Var and EnKF configurations that are normally used for chemical data assimilation applications. Both data assimilation systems have online estimation of error variances based on the Desroziers' method and share the same correlation model for all prescribed error correlations (i.e. the background error covariance for 4D-Var, initial error and model error for EnKF) so that each data assimilation system is nearly optimal and can also be compared to each other. A previous comparison study by (Skachko et al. 2014) showed that for chemical tracer transport only both assimilation system provide results of essentially similar quality despite the difference in practical implementation of each method: the 4D-Var was applied in its strong constraint formulation with a 24 h assimilation window with the assumption of no model error over this period, whereas the EnKF was used to sequentially assimilate observations every 30 minutes with model error perturbations added every 30 minutes."

Then the following text is added at the end of our conclusions: "Another possibilities may be considered to properly compare two essentially different data assimilation systems. First, a 4D-EnKF approach, where model error perturbations only applied at the beginning of each 4D-Var assimilation window to be equivalent with a strong-constraint 4D-Var, may be considered. This would allow a longer assimilation window to be used for the EnKF. In this case, the analysis would be forced to simultaneously fit all of the observations distributed over a longer window, while still satisfying the model equations, as in 4D-Var. Second, the use of a weak-constraint 4D-Var including model error would also reduce the differences between two considered approaches. "

Specific comments:

In response to the third specific comment, your revised sentence seems imprecise: "For comparison purposes, we apply the same estimate procedure in the 4D-Var data

assimilation, where both, the background and observation error covariance matrices are estimated using the Desroziers' method." I presume it is only the scale factors for both covariance matrices that are estimated and not the full matrices? Please improve the wording.

The sentence is now written as: "For comparison purposes, we apply the same estimate procedure in the 4D-Var data assimilation, where both scale factors of the background and observation error covariance matrices are estimated using the Desroziers' method."

In response to the fifth specific comment, your revised sentence does not clear up my concern: "The second issue in EnKF with comprehensive atmospheric chemistry models is the spurious error, that occurs when species are weakly chemically related at the same location." The term "spurious error" is very ambiguous... how can error be spurious? I believe this is again where "error" is used in place of "error covariance". Only the "estimated error covariance" is spurious. [The word "error" on its own really should be reserved for the difference between an estimate and the truth and I don't think this is what is meant in this case. I realize that some published papers have used "error" to mean "error standard deviation" or "error covariance", but I believe this has needlessly caused confusion for some people in the DA community.]

The sentence is rewritten as follows: "The second issue in EnKF with comprehensive atmospheric chemistry models is the noise in the cross-covariance between species, that occurs when species are weakly chemically related at the same location."

**References**

Skachko, S., Errera, Q., Ménard, R., Christophe, Y., and Chabrillat, S.: Comparison of the ensemble Kalman filter and 4D-Var assimilation methods using a stratospheric tracer transport model, Geosci. Model Dev., 7, 1451-1465, doi:10.5194/gmd-7-1451-2014, 2014.

---

## Short Comment (SC1) · 6 Jun 2016

Dear authors,

In my role as Executive editor of GMD, I would like to bring to your attention our Editorial version 1.1:

http://www.geosci-model-dev.net/8/3487/2015/gmd-8-3487-2015.html

This highlights some requirements of papers published in GMD, which is also available on the GMD website in the 'Manuscript Types' section:

http://www.geoscientific-model-development.net/submission/manuscript_types.html

In particular, please note that for your paper, the following requirements have not been met in the Discussions paper:

[Figure]

- "The main paper must give the model name and version number (or other unique identifier) in the title."

- "If the model development relates to a single model then the model name and the version number must be included in the title of the paper. If the main intention of an article is to make a general (i.e. model independent) statement about the usefulness of a new development, but the usefulness is shown with the help of one specific model, the model name and version number must be stated in the title. The title could have a form such as, "Title outlining amazing generic advance: a case study with Model XXX (version Y)"."

Please correct this in your revised submission to GMD.

Yours,

Astrid Kerkweg

---

## Author Comment (AC4) · 6 Jun 2016

We thank the referee for the fruitful discussion of the paper. The proposed text has been taken into account as is.

---

## Author Comment (AC5) · 6 Jun 2016

June 6, 2016

The author's responses are marked in blue. We would like to thank the anonymous referee 3 to the useful remarks.

This study compares EnKF and 4D-Var data assimilation methods applied to a chemistry transport model. The purpose is to compare relative merits of the two methods on long time (short windows) atmospheric chemistry data assimilation with prescribed flow fields.

Major comments:

1 EnKF Experimental setup: Page 6: "the model error term is added to observed species only." What is the rationale for this?

Perturbing all 58 species of the model state vector results in the noisy cross-species error covariances. A simplified example of such set-up (where the cross-covariances between the ozone and $N_2O$ only) is shown in the experiment EnKF-CC. When non-observed species are not (or weakly) chemically related with the observed species, the noise introduced to the EnKF error covariances essentially

The same L operator seems to be used both for 4D-Var and EnKF, but at lest in the definition of $\eta$ in (7), and (1) or (2), L lives in different spaces.

The operator $\mathbf{L}$ is defined by Eq. (3) for both, 4D-Var and EnKF systems in the spectral

space. The algorithm to generate EnKF state perturbations is then identical to the algorithm of the 4D-Var background error covariance generation. However, the operator $\mathbf{L}$ is applied to the normally distributed random deviate $\zeta_i$ (Eq. (6)) rather than to the control variable $\xi$ (Eq. (1))

The authors claim that the same error covariances are used in both cases [page 14: "the same correlation model for all prescribed error correlations (i.e. the background error for 4D-Var, initial error and model error for EnKF)"]; however, on page 8 around line 10, they seem to indicate different localization operators that come in to build B. This should be clarified.

We have given in the manuscript a reference to our previous study, where it had been explained in more details: "The EnKF uses as localization method a Schur product with a compact support correlation function. The use of Schur product reduces the resulting correlation length scales. In order to maintain the correlations of the EnKF analysis comparable to those of the 4D-Var system, a different setting of the correlation length scales is adopted to generate the model error. Let $\mathbf{C}$ be a matrix resulting from the Schur product of two matrices $\mathbf{A}$ and $\mathbf{B}$: $\mathbf{C} = \mathbf{A} \circ \mathbf{B}$. If the correlation length scales of $\mathbf{A}$ and $\mathbf{B}$ are, respectively $L_A$ and $L_B$, the correlation length scale of $\mathbf{C}$ is given by Gaspari and Cohn, (1999):

$$\frac{1}{L_C^2} = \frac{1}{L_A^2} + \frac{1}{L_B^2}. \tag{1}$$

In our case, $L_A$ corresponds to the correlation length scale $L_{loc}$ of the compact support correlation function $\rho$ and $L_B$ corresponds to the correlation length scale of the forecast ensemble covariance matrix $\mathbf{B}_e$, denoted in the following by $L_e$. Similarly, $L_C$ corresponds to the correlation length scale of the analysis ensemble covariance matrix, denoted in the following by the effective correlation length scale $L_{eff}$. As we would like to maintain the $L_{eff}$ equal to the Gaussian correlation length scales used in the 4D-Var (i.e. $L_0^h$=800 km and $L_0^v$=1 level), we need to set $L_{loc}$ and $L_e$ such that $L_{eff} = L_0$. "

Cross species localization: In Section 5 the authors discuss the effects of interspecies

localization. It is unclear to me what is done here. Is ENFK-CC the same as EnKF except that in EnKF-CC the O3 and NO2 are localized? If that is the case, then this is problematic because one cannot choose to localize some species and not localize the others because it introduces transients that may lead to spurious bias oscillations. This should be clarified as well.

The experiment, denoted as EnKF-CC, involves the cross-species error covariance between two weakly chemically related species, the ozone and $N_2O$. In other words, we consider the assimilation test, where the observational updates of ozone are obtained using both the ozone and $N_2O$ measurements, and the updates of $N_2O$ are also computed using both datasets. This is done in addition to the procedure of spatial localization that has been applied to all observed species.

Minor comment:

Page 4 line 2: "cross-covariance between species are taken into account automatically using the 4D-Var adjoint mode" is not clear to me. How is this achieved?

In the 4D-Var approach, we have a direct and an adjoint chemistry model. The chemistry model includes all possible chemical interactions between species. Hence, the 4D-Var computes the observational updates of all model state variables (observed and non-observed) from all available observations.

**References**

Gaspari, G. and Cohn, S. E.: Construction of correlation functions in two and three dimensions, Q. J. R. Meteorol. Soc., 125, 723–757, 1999.

---

## Author Comment (AC6) · 6 Jun 2016

Dear Dr. Kerkweg, the title of our article is changed accordingly: "EnKF and 4D-Var Data Assimilation with the Chemistry Transport Model BASCOE (version 05.06)".

Thank you for your comment. Sincerely yours, Sergey Skachko